# On the Unreasonable Effectiveness
# of Last-layer Retraining

**John Collins Hill**                                                      *jhill326@gatech.edu*
*School of Electrical and Computer Engineering*
*Georgia Institute of Technology*

**Tyler LaBonte**                                                         *tlabonte@gatech.edu*
*H. Milton Stewart School of Industrial and Systems Engineering*
*Georgia Institute of Technology*

**Xinchen Zhang**                                                      *xzhang941@gatech.edu*
*School of Electrical and Computer Engineering*
*Georgia Institute of Technology*

**Vidya Muthukumar**                                              *vmuthukumar8@gatech.edu*
*H. Milton Stewart School of Industrial and Systems Engineering,*
*School of Electrical and Computer Engineering*
*Georgia Institute of Technology*

**Reviewed on OpenReview:** *https://openreview.net/forum?id=h81ztbrkFb*

## Abstract

Last-layer retraining (LLR) methods — wherein the last layer of a neural network is reinitialized and retrained on a held-out set following ERM training — have garnered interest as an efficient approach to rectify dependence on spurious correlations and improve performance on minority groups. Surprisingly, LLR has been found to improve worst-group accuracy even when the held-out set is an imbalanced subset of the training set. We initially hypothesize that this "unreasonable effectiveness" of LLR is explained by its ability to mitigate neural collapse through the held-out set, resulting in the implicit bias of gradient descent benefiting robustness. Our empirical investigation *does not* support this hypothesis. Instead, we present strong evidence for an alternative hypothesis: that the success of LLR is primarily due to better group balance in the held-out set. We conclude by showing how the recent algorithms CB-LLR and AFR perform implicit group-balancing to elicit a robustness improvement.

## 1 Introduction

The standard neural network training procedure of empirical risk minimization (ERM) (Vapnik, 1998), which minimizes the average classification loss, is well-known to overfit to spurious correlations in the training set (Geirhos et al., 2020). The conjunction of target labels and spurious features form *groups* within the dataset; the smallest groups in each class, termed *minority groups*, are often most difficult to correctly classify at test time (Oren et al., 2019). While ERM may exhibit high average accuracy on these datasets, performance on minority groups can be no better than random guessing (Shah et al., 2020). Therefore, when robust minority group performance is critical, worst-group accuracy (WGA) is often a more useful metric. Due to the prevalence of spurious features and minority groups (race, gender, age, *etc.*) in high consequence applications such as medicine (Zech et al., 2018) and criminal justice (Chouldechova, 2016), significant work has focused on algorithms which maximize WGA — also called group robustness (Sagawa et al., 2020a).

A promising family of group robustness methods are based on last-layer retraining (LLR), a family of interventions wherein the last layer is reinitialized and retrained on a held-out set following ERM training. The original LLR method, called deep feature reweighting (DFR), requires the held-out set to comprise an equal amount of data from each group (Izmailov et al., 2022; Kirichenko et al., 2023). This limits its practical application, as the groups are often unknown ahead of time or difficult to annotate.[1] While DFR still performs the best, group information (even on the validation set) was recently found to be unnecessary to observe *some* non-trivial gain in WGA over ERM if the validation set is class-balanced (Qiu et al., 2023; LaBonte et al., 2023). In particular, LaBonte et al. (2023) find that LLR on a held-out *i.i.d.* subset of the training set — suffering from the same group imbalance as the training set — is sufficient for improved WGA. This surprising observation led LaBonte et al. (2023) to term LLR a "free lunch" for group robustness, but why this "free lunch" arose in the first place remained unclear.

**Contributions.** In this paper, we take a "scientific method" approach to investigate why LLR on an imbalanced held-out subset of the training set can perform so well. We make the following main contributions:

- We propose an initial hypothesis, visualized in Figure 1, that neural collapse on the training set results in a biased ERM classifier since the class means are dominated by majority group data. Since features are *not* collapsed on the held-out set (Hui et al., 2022), this would lead to varying behavior on majority and minority group data. Our hypothesis is that the implicit bias of gradient descent during LLR then elicits a *maximum-margin linear classifier* on the features, which has been connected in some settings to better robustness guarantees (Chaudhuri et al., 2023).

- We put forth evidence which *does not* support our initial hypothesis. In particular, we show that neural collapse does not seem to occur during the standard number of epochs on four benchmark datasets. Moreover, we show that convergence of the LLR classifier to the maximum-margin solution is extremely slow. Since computation of the neural collapse metric $\mathcal{NC}_1$ was previously infeasible for large-scale models, we propose a memory-efficient algorithm to compute a stochastic estimate of $\mathcal{NC}_1$ via the Hutchinson trace estimator (Hutchinson, 1989).

- We present strong evidence for an alternative hypothesis: that the success of LLR is primarily due to *better group balance.*[2]e define *better group balance* with respect to the uniform distribution over groups. This may not induce the highest possible WGA via last-layer retraining, though it is often close (Qiao et al., 2025). Our experiments indicate that LLR does not improve over ERM when the held-out set has the same group balance as the training set. On the other hand, LLR with a better group balance enjoys drastically improved WGA, and vice versa. We show that the test WGA of LLR is highly correlated with the test WGA of ERM when controlling for data quantity and group balance, with Pearson correlation coefficient $r > 0.9$ on Waterbirds and CivilComments.

- Ultimately, we reevaluate the "free lunch" interpretation of LLR by showing that class balanced LLR (CB-LLR) (LaBonte et al., 2023) and automatic feature reweighting (AFR) (Qiu et al., 2023) owe their improved WGA primarily to *implicit group-balancing* on the held-out set. On a positive note, we show that LLR can recover the WGA of an optimally class-balanced model even when ERM was not optimally class-balanced. More broadly, LLR remains an effective method to achieve group robustness using group annotations only on the held-out set (or even proxies thereof, such as in SELF (LaBonte et al., 2023) and AFR (Qiu et al., 2023)). When group annotations are expensive, LLR on a more group-balanced held-out set is therefore an attractive improvement over ERM.

## 1.1 Related work

**Spurious correlations.** Reliance on spurious correlations is a widely-studied phenomenon in machine learning which is known to exacerbate bias and hinder generalization (see, *e.g.,* Beery et al. (2018); Geirhos

---

[1]However, DFR compares favorably in this respect to methods which additionally require group annotations for the entire training set, such as group distributionally robust optimization (DRO) (Sagawa et al., 2020a).
[2]W

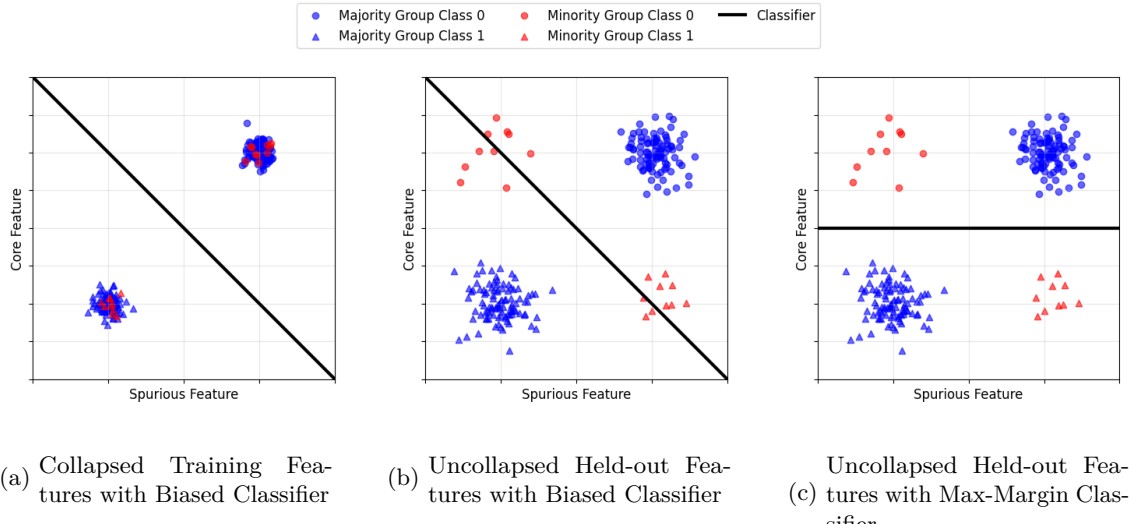

(a) Collapsed Training Features with Biased Classifier

(b) Uncollapsed Held-out Features with Biased Classifier

(c) Uncollapsed Held-out Features with Max-Margin Classifier

Figure 1: **Visualization of our initial hypothesis.** In Figure 1(a), we represent training set features collapsing to their class means, dominated by majority group data. The resulting ERM classifier is biased on unseen data, *i.e.,* the LLR held-out set or the test set. Figure 1(b) shows how the biased classifier performs poorly on uncollapsed minority group features while still successfully classifying uncollapsed majority points. In Figure 1(c), we present the final step of our hypothesis. During LLR, the features are not collapsed on the held-out set, and so the implicit bias of gradient descent could elicit a maximum-margin classifier which is invariant to the spurious feature. Importantly, our empirical investigation *does not* support this hypothesis. Instead, we find that the success of LLR is primarily explained by better group balance in the held-out set.

et al. (2020); Xiao et al. (2021)). Much work has been devoted to mitigating the adverse affects from spurious correlations (Sagawa et al., 2020a). Methods which have been demonstrated to reduce the bias caused by spurious correlations without the need for large quantities of group labels, such as Just Train Twice (Liu et al., 2021a), Deep Feature Reweighting (Kirichenko et al., 2023), Classifier Retraining on Independent Splits (Nguyen et al., 2023), and class-balancing (Idrissi et al., 2022; Chaudhuri et al., 2023; Shwartz-Ziv et al., 2023; LaBonte et al., 2024), are of particular interest. While an understanding of why shortcut learning occurs is still incomplete (Izmailov et al., 2022), recent studies are beginning to unravel the joint dynamics of core and spurious feature learning (Qiu et al., 2024).

**Last-layer retraining (LLR).** Proposed as Deep Feature Reweighting by Izmailov et al. (2022); Kirichenko et al. (2023), LLR has recently garnered interest as an efficient approach to improve group robustness in a variety of settings (Qiu et al., 2023; LaBonte et al., 2023; Stromberg et al., 2024; Park et al., 2025; Qiao et al., 2025). The success of LLR is limited when the held-out set has the same (im)balance as the training set, but more pronounced when the held-out set if perfectly group-balanced (as in DFR (Izmailov et al., 2022; Kirichenko et al., 2023) or even class-balanced (Qiu et al., 2023; LaBonte et al., 2023). In all cases, the success of LLR is predicated on ERM learning both core and spurious features during training, but relying too heavily on spurious features within the last layer (Kirichenko et al., 2023; Ye et al., 2023). Practically, LLR is appealing because of its ability to achieve much of the benefit of full fine-tuning at a fraction of the computational cost while only requiring limited use of group annotations.

**Neural collapse.** We study how LLR interacts with the phenomenon of *neural collapse*, first identified by Papyan et al. (2020); Han et al. (2022). Neural collapse describes the collapse of *feature variability* during training — in other words, the features of each datapoint within a given class collapse to the mean feature vector inside that class. Recently, neural collapse has been studied in connection with fairness and spurious correlations in the context of ERM by Lu et al. (2024); Wang et al. (2024); Chen et al. (2024); Xu et al. (2025). Additionally, investigations of neural collapse in imbalanced data settings (though mostly focused on minority classes) (Fang et al., 2021; Thrampoulidis et al., 2022; Dang et al., 2023; Hong & Ling, 2023)

provide motivation for studying the impact of minority group collapse. The role neural collapse plays in group robust ERM performance is still unclear. Neural collapse has been observed to produce less biased (Xu et al., 2025), but also less separable group representations (Lu et al., 2024). Importantly for our purposes, neural collapse has been classified as an *optimization phenomenon*, meaning it occurs only on the training set and not on held-out data (Hui et al., 2022). Our initial hypothesis was that this difference in feature behavior between seen and unseen data could help explain why ERM underperforms LLR (due to the fact that LLR has additional access to the unseen held-out set as a result of its two-stage training protocol). Ultimately, we disprove this hypothesis.

**Implicit bias and margin.**   The second stage of LLR is simply logistic regression performed on the learned ERM features (Kirichenko et al., 2023). Therefore, we expect implicit bias results for logistic regression to apply to our setting. We draw on the literature studying the implicit bias of gradient descent in logistic regression toward the maximum $\ell_2$-margin classifier, studied in the separable case by Soudry et al. (2018) and in the non-separable case by Ji & Telgarsky (2019). Large minimum margin over the training data (which is the default definition of margin) can imply good generalization upper bounds (Bartlett et al., 1998; Koltchinskii & Panchenko, 2002). Recent results suggest that, under certain conditions, the maximum-margin classifier coincides with the classifier achieving optimal worst-group accuracy (Chaudhuri et al., 2023), offering a possible bridge between implicit-bias theory and robust-optimization perspectives on group fairness.

## 2   Preliminaries

**Problem Setup.**   We study classification with input domain $\mathcal{X}$ and label set $\mathcal{Y}$ in the presence of *spurious features* $\mathcal{S}$, where each example $x \in \mathcal{X}$ is associated with exactly one spurious feature $s(x) \in \mathcal{S}$. The training, validation, and test datasets are partitioned into *groups* $\mathcal{G} \coloneqq \mathcal{Y} \times \mathcal{S}$; let $\Omega_g \subseteq \{1, \ldots, m\}$ and $\Omega_y \subseteq \{1, \ldots, m\}$ denote the indices of training points in group $g \in \mathcal{G}$ and class $y \in \mathcal{Y}$, respectively. We call groups with maximal $|\Omega_g|$ in the training set the *majority* groups and the rest *minority* groups; the *worst* group(s) are those with the lowest test accuracy. (In general, the majority and minority groups will not vary between the training and test sets.) Our objective is to learn models that perform uniformly well across groups despite imbalance, quantified by *worst-group accuracy* (WGA) — the minimum accuracy over all groups in $\mathcal{G}$ (Sagawa et al., 2020a).

**Class-balancing.**   We adopt the definition of a class-balanced dataset as one where the expected number of samples from every class in the label space $\mathcal{Y}$ is identical. Adhering to the framework established by LaBonte et al. (2024), we evaluate three distinct balancing strategies: *subsetting*, *upsampling*, and *upweighting*.

- *Subsetting*: This method enforces balance by strictly limiting the size of all classes to match that of the minority class. We achieve this by uniformly subsampling the majority classes without replacement. It is important to note that this subset is generated once prior to training and remains static throughout the optimization process.

- *Upsampling*: This approach utilizes the full dataset but modifies the sampling distribution. To ensure that mini-batches are class-balanced in expectation, we adopt a two-step generative process for drawing a training example: first, we sample a label $y \sim \text{Unif}(\mathcal{Y})$, and subsequently sample an input $x$ from the class-conditional empirical distribution $\hat{p}(\cdot \mid y)$.

- *Upweighting*: Rather than altering the data distribution, this method modifies the objective function. We define the class-imbalance ratio, $\gamma$, as the ratio of the majority class size to the minority class size. We then scale the contribution of minority samples to the loss by $\gamma$. Formally, for a model $f$, input $x$, and label $y$, the modified loss function is defined by $\mathcal{L}_{\text{balanced}}(f(x), y) = w_y \cdot \ell(f(x), y)$. Here, $w_y = \gamma$ if $y$ belongs to the minority class, and $w_y = 1$ otherwise. Upweighting is mathematically equivalent to upsampling in expectation over the sampling probabilities.

**Datasets.**   We evaluate our approach on four standard benchmarks: Waterbirds (Welinder et al., 2010; Wah et al., 2011; Sagawa et al., 2020a), CelebA (Liu et al., 2015; Sagawa et al., 2020a), CivilComments (Borkan

et al., 2019; Koh et al., 2021), and MultiNLI (Williams et al., 2018; Sagawa et al., 2020a). For each dataset, we define the input $x$, the target label $y$, and the spurious attribute $a$. Note that Waterbirds is the only dataset exhibiting a distribution shift between training and held-out sets, while MultiNLI is the only dataset that is class-balanced *a priori*.

- *Waterbirds* (Welinder et al., 2010; Wah et al., 2011; Sagawa et al., 2020a): $x$ consists of bird images classified by species into $y \in \{\text{landbird}, \text{waterbird}\}$. The spurious feature $a$ represents the background environment (land vs. water), where backgrounds are highly correlated with their corresponding species classification.[3]

- *CelebA* (Liu et al., 2015; Sagawa et al., 2020a): $x$ consists of celebrity face images where the target $y$ is hair color (blond vs. non-blond). The spurious attribute $a$ is gender; specifically, the training distribution is skewed such that the intersectional group of blond males is significantly underrepresented.

- *CivilComments* (Borkan et al., 2019; Koh et al., 2021): $x$ comprises online comments where the target $y$ is toxicity. The spurious feature $a$ captures the presence of demographic identities (e.g., race, gender, religion). Following Sagawa et al. (2020a); Idrissi et al. (2022); Izmailov et al. (2022); Kirichenko et al. (2023); LaBonte et al. (2023), we aggregate these identity categories into a single feature $a$, which is correlated with the toxicity label $y$.[4]

- *MultiNLI* (Williams et al., 2018; Sagawa et al., 2020a): $x$ consists of sentence pairs for natural language inference where $y \in \{\text{contradiction}, \text{entailment}, \text{neutral}\}$. The spurious feature $a$ denotes the presence of negation words in the second sentence; these negations are spuriously correlated with the contradiction class.

For Waterbirds and CelebA we fine-tune ResNet-50 (He et al., 2016) pretrained on ImageNet-1K (Russakovsky et al., 2015); for CivilComments and MultiNLI we use BERT-Base (Devlin et al., 2019) pretrained on BookCorpus and English Wikipedia (Zhu et al., 2015). Our last-layer retraining (LLR) implementation uses SGD without scheduling or explicit regularization. Following common practice, the held-out set is 20% of the training data for Waterbirds and half of the validation set for the other datasets (Kirichenko et al., 2023; Qiu et al., 2023; LaBonte et al., 2023). Dataset specifics and training details appear in Appendices B.1 and B.2.

**Methods.** We adopt a unified terminology for the family of two-stage retraining methods we study:

- *Last-layer Retraining (LLR)* refers to any procedure that *(i)* trains a feature extractor using standard ERM on the training set, and *(ii)* reinitializes and retrains only the final linear layer on a held-out dataset not seen during the first stage. Importantly, generic LLR makes *no assumption* about the distribution of this held-out data—it may be arbitrarily imbalanced across classes or groups.

- *Deep Feature Reweighting (DFR)* (Izmailov et al., 2022; Kirichenko et al., 2023) is a *specific instance* of LLR in which the held-out dataset is explicitly *group-balanced*: each group $g \in \mathcal{G}$ is equally represented. In practice, DFR freezes the ERM-trained feature extractor, constructs a subsampled group-balanced held-out set (requiring access to group annotations), and fits a new linear classifier on this held-out set. DFR can thus be viewed as an "idealized" variant of LLR designed to directly improve worst-group accuracy.

- *Class-balanced Last-layer Retraining (CB-LLR)* (LaBonte et al., 2023) denotes another LLR variant where the held-out data are balanced across *classes* rather than groups. While class proportions are equalized, group imbalance within each class often remains.

---

[3]The Waterbirds dataset is known to contain incorrect labels (Taghanaki et al., 2022). We utilize the standard uncorrected labels for all of our results.

[4]There exists another version of CivilComments (used by Liu et al. (2021a); Zhang et al. (2022); Qiu et al. (2023)) where identity categories are not merged into one spurious feature. Both versions of CivilComments use the WILDS split (Koh et al., 2021).

- *Automatic Feature Reweighting (AFR)* (Qiu et al., 2023) refers to the LLR variant in which the held-out set is drawn from the same distribution as the training data, but the last layer is retrained using a *weighted loss function* intended to implicitly encourage group balance. The weighted loss function in AFR prioritizes points upon which the ERM model performs poorly. In particular, AFR introduces a weight $\omega_i$ for each held-out example $i$ as follows:

$$\omega_i = \frac{\beta_{y_i} \exp(-\gamma \hat{p}_i)}{\sum_j^M \beta_{y_j} \exp(-\gamma \hat{p}_j)}, \tag{1}$$

  where $\beta_y$ is one divided by the number of examples belonging to class $y$ in the held-out set, $M$ is the total number of points in the held-out set, $\hat{p}_i$ is the probability for the correct class $y_i$ and $\gamma \geq 0$ is a tunable inverse temperature parameter.

## 3 Neural collapse and implicit bias do not explain LLR

The interpretation of LLR methods as logistic regression on convergent ERM features lends itself to a possible understanding combining a nonlinear model training phenomenon (*neural collapse*) and a logistic regression phenomenon (*implicit bias*). We focus on the weakest measure of the neural collapse phenomenon, namely that the penultimate layer features collapse to their class means over the course of ERM training (Papyan et al., 2020). Importantly, neural collapse has been described as an *optimization phenomenon*, occurring only on the training set and not on held-out data (Hui et al., 2022). Therefore, while all group features collapse to their class means on the training set, groups will not be collapsed in the held-out set. On the other hand, implicit bias results state that linear classifiers trained via gradient descent with the unregularized logistic loss — like the LLR classifier on the held-out set — converge in direction to the maximum-margin SVM solution (Soudry et al., 2018; Ji & Telgarsky, 2019).

Combining these two ideas, we developed the following initial hypothesis (visualized in Figure 1). We hypothesized that the model undergoes neural collapse during ERM training, causing the features to collapse to their class means. These class means, however, are dominated by the largest groups in each class — resulting in ERM learning a biased classifier. Nevertheless, during LLR, the features are not collapsed as the held-out set was not seen during ERM (Hui et al., 2022). Since LLR learns a linear classifier on the held-out set features, the implicit bias of gradient descent would then elicit a maximum-margin classifier that might enjoy better robustness guarantees (see, e.g. (Chaudhuri et al., 2023)).

In this section, we present evidence which *does not* support our initial hypothesis. Specifically, we find that neural collapse either does not occur or occurs after the standard number of epochs of ERM training. Moreover, convergence of the LLR classifier to the maximum-margin solution is extremely slow, and the minimum $\ell_2$-margin over the training data is not well-correlated with group accuracy.

### 3.1 Neural collapse may not occur during standard ERM training

We will measure the collapse of variability among class representations via a metric called $\mathcal{NC}_1$ (Han et al., 2022). For each class $y \in \mathcal{Y}$ and each training example $i \in \Omega_y$ in that class, we denote the penultimate layer feature vector of $i$ as $f_{y,i}$. Neural collapse posits that features collapse to their respective class means

$$\mu_y := \frac{1}{|\Omega_y|} \sum_{i \in \Omega_y} f_{y,i}.$$

We compute an empirical metric of this collapse using the intra-class covariance matrix $\Sigma_A := \frac{1}{m} \sum_{y \in \mathcal{Y}} \sum_{i \in \Omega_y} (f_i - \mu_y)(f_i - \mu_y)^\top$, the global feature mean $\mu_G := \frac{1}{|\mathcal{Y}|} \sum_{y \in \mathcal{Y}} \mu_y$, and the inter-class covariance matrix $\Sigma_R := \frac{1}{|\mathcal{Y}|} \sum_{y \in \mathcal{Y}} (\mu_y - \mu_G)(\mu_y - \mu_G)^\top$. We then define

$$\mathcal{NC}_1 := \frac{1}{|\mathcal{Y}|} \text{tr}(\Sigma_A \Sigma_R^\dagger),$$

---

**Algorithm 1 Memory efficient computation of $\mathcal{NC}_1$ estimate.** Our method combines the Hutchinson trace estimator (Hutchinson, 1989) with efficient matrix-vector products for $\Sigma_A$ and $\Sigma_R$ to estimate $\mathcal{NC}_1$ in $O(N)$ memory, instead of the $O(N^2)$ memory required to compute $\mathcal{NC}_1$ exactly.

---

**Require:** Number of samples $K$, access to matrix–vector products $\Sigma_A x$ and $\Sigma_R x$.
**Ensure:** Approximation $\widehat{\mathcal{NC}}_1 \approx \frac{1}{|\mathcal{Y}|}\mathrm{tr}(\Sigma_A \Sigma_R^\dagger)$.

1: **for** $j = 1, \ldots, K$ **do**
2:      Sample random vector $z_j \sim \mathcal{N}(0, I_N)$          $\triangleright$ Or Rademacher $\{\pm 1\}^N$
3:      Solve $\Sigma_R x_j = z_j$ for $x_j$ using an iterative solver          $\triangleright$ $x_j \approx \Sigma_R^\dagger z_j$
4:      Compute $y_j \leftarrow \Sigma_A x_j$
5:      Compute scalar estimate $s_j \leftarrow z_j^\top y_j$
6: **end for**
7: **return** $\widehat{\mathcal{NC}}_1 \leftarrow \frac{1}{K|\mathcal{Y}|} \sum_{j=1}^{K} s_j$

---

where $\Sigma_R^\dagger$ denotes the pseudo-inverse of $\Sigma_R$. Neural collapse is formalized by $\mathcal{NC}_1 \to 0$ (Han et al., 2022; Kothapalli, 2023).[5]

**Algorithm Design.** Computing the neural collapse metric $\mathcal{NC}_1$ exactly requires constructing and storing the matrices $\Sigma_A$ and $\Sigma_R$, each of dimension $N \times N$, where $N$ is the feature dimension of the penultimate layer. For modern neural networks, storing these matrices explicitly is memory-prohibitive. For example, in the case of BERT-Base trained on CivilComments, $N$ exceeds 150,000. To address this issue, we propose a memory-efficient approximation based on the *Hutchinson trace estimator* (Hutchinson, 1989).

The Hutchinson estimator provides a stochastic approximation for the trace of a matrix:

$$\mathrm{tr}(A) = \mathbb{E}[z^\top A z] \approx \frac{1}{K} \sum_{i=1}^{K} z_i^\top A z_i,$$

where $A \in \mathbb{R}^{N \times N}$, each random vector $z_i \in \mathbb{R}^N$ satisfies $\mathbb{E}[z_i z_i^\top] = I$, and $K$ is the number of random samples used to approximate the expectation. Typically, $z_i$ is drawn from a Rademacher or standard normal distribution, but any isotropic distribution suffices.

In our setting, we wish to approximate $\mathcal{NC}_1$ without ever explicitly forming $\Sigma_A$ or $\Sigma_R$. Note that both matrices admit efficient matrix–vector products:

$$\Sigma_A x = \frac{1}{m} \sum_{y \in \mathcal{Y}} \sum_{i \in \Omega_y} (f_i - \mu_y)\big((f_i - \mu_y)^\top x\big),$$

$$\Sigma_R x = \frac{1}{|\mathcal{Y}|} \sum_{y \in \mathcal{Y}} (\mu_y - \mu_G)\big((\mu_y - \mu_G)^\top x\big),$$

where $\mu_y$ is the class mean and $\mu_G$ the global mean. These expressions can be computed by streaming over the data, requiring only a constant number of $N$-dimensional vectors in memory. Combining these observations yields a stochastic algorithm (Algorithm 1) capable of estimating $\mathcal{NC}_1$ with only $O(N)$ memory, compared to the $O(N^2)$ memory required to compute $\mathcal{NC}_1$ exactly. We compare the precise memory requirements of both methods in Table 1.

This procedure introduces two sources of error:

1. *Linear solve error*: Each iterative solve of $\Sigma_R x_j = z_j$ is performed to finite precision.

2. *Monte Carlo error*: The Hutchinson estimator uses finite samples $K$ to approximate an expectation.

Both sources of error can be reduced arbitrarily by increasing computation time. Notably, the variance of the Hutchinson estimator decreases with the feature dimension $N$ (Hutchinson, 1989). We show in Appendix C (Proposition 1) that the relative variance of $\widehat{\mathcal{NC}}_1$ scales as $O(1/KN)$, so fewer samples $K$ are required for larger models, corresponding to our regime of primary interest.

---

[5] A slightly less precise formulation $\Sigma_A \to 0$ was studied by Papyan et al. (2020).

Table 1: **Computing a stochastic estimate of $\mathcal{NC}_1$ using Algorithm 1 is drastically more memory efficient than computing $\mathcal{NC}_1$ exactly.** We display the vectorized feature dimensions $N$ for ResNet-50 on the vision datasets and BERT-Base on the language datasets. Exact computation $\mathcal{NC}_1$ requires storing two $N \times N$ matrices of double-precision floating-point numbers, while our estimate of $\mathcal{NC}_1$ only requires storing three $N$ dimensional vectors of double-precision floating-point numbers. We see that our stochastic estimate of $\mathcal{NC}_1$ requires many orders of magnitude less memory.

| Dataset | Waterbirds | CelebA | CivilComments | MultiNLI |
|---|---|---|---|---|
| Vectorized Feature Dimension $N$ | 100,352 | 100,352 | 168,960 | 98,304 |
| Memory Requirement of Exact $\mathcal{NC}_1$ Computation | 150.06 GiB | 150.06 GiB | 425.4 GiB | 144 GiB |
| Memory Requirement of Algorithm 1 | 2.297 MiB | 2.297 MiB | 3.867 MiB | 2.25 MiB |

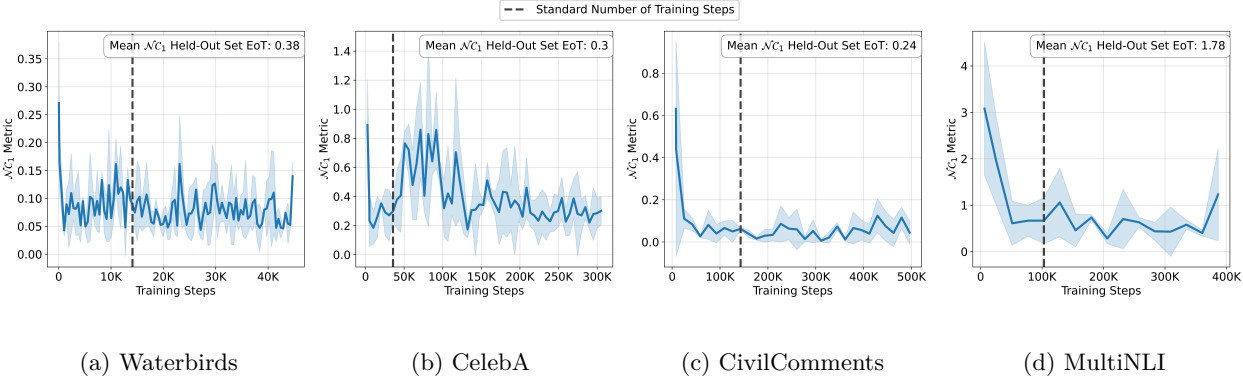

(a) Waterbirds      (b) CelebA      (c) CivilComments      (d) MultiNLI

Figure 2: **Collapse of class feature variability occurs after standard ERM training, if at all.** We plot a stochastic estimate of the empirical metric of neural collapse $\mathcal{NC}_1$ using Algorithm 1 throughout the training run of a ResNet-50 on Waterbirds and CelebA and a BERT-Base on CivilComments and MultiNLI. For Waterbirds and CelebA, $\mathcal{NC}_1$ is computed using $K = 10$ random vectors, while for CivilComments and MultiNLI, $\mathcal{NC}_1$ is computed using $K = 3$ random vectors. Each plot displays the mean and standard deviation for $\mathcal{NC}_1$ computed across 3 experimental seeds. We also display the mean $\mathcal{NC}_1$ metric computed on the features of the held-out set at the end of training (EoT).

**Results.** If neural collapse plays a significant role in ERM learning a biased classifier prior to LLR, then a collapse in feature variability should be observed during a standard number of ERM training steps. We display $\mathcal{NC}_1$ for a ResNet-50 on Waterbirds and CelebA and for a BERT-Base for CivilComments and MultiNLI in Figure 2. We train on both datasets for much longer than normal and denote the standard number of training steps in the spurious correlations literature (Sagawa et al., 2020a; Kirichenko et al., 2023) as dotted vertical lines in Figure 2.

In Figure 2, we do not see strong evidence that neural collapse occurs on any dataset during the standard number of training steps. In fact, even after training more than $3\times$ the standard number of steps we still do not see $\mathcal{NC}_1 \to 0$. Beyond an initial decrease in the first few epochs of training, $\mathcal{NC}_1$ computed on the training set remains relatively constant and comparable to $\mathcal{NC}_1$ computed on the held-out. For all datasets, $\mathcal{NC}_1$ computed on the training sets and $\mathcal{NC}_1$ computed on the held-out sets are within an order of magnitude. Thus, it is unlikely that neural collapse significantly influences the behavior of the ERM classifier prior to LLR in standard group robustness benchmarks and training protocols.

## 3.2 The minimum margin of LLR is not predictive of robust generalization

In classical learning theory, maximizing minimum $\ell_2$ training margin is well understood to provide good generalization upper bounds (Bartlett et al., 1998; Koltchinskii & Panchenko, 2002). Therefore, assuming

Table 2: **Group test accuracy is not well-correlated with the minimum $\ell_2$ margin on the held-out set.** We compute the Pearson correlation coefficient between the test group accuracy of a ResNet-50 (BERT-Base) trained on Waterbirds and CelebA (CivilComments and MultiNLI) and the minimum margin of each group in the held-out set across 3 experimental seeds. We see that held-out set minimum $\ell_2$ margin is not well correlated with test accuracy. A complete breakdown of each dataset including distribution and group sizes is included in Table 7.

| Dataset | Group 0 | Group 1 | Group 2 | Group 3 | Group 4 | Group 5 |
|---|---|---|---|---|---|---|
| Waterbirds | 0.279 | 0.285 | 0.288 | 0.266 | — | — |
| CelebA | 0.526 | 0.473 | 0.458 | 0.478 | — | — |
| CivilComments | -0.074 | -0.143 | -0.008 | 0.055 | — | — |
| MultiNLI | 0.303 | 0.497 | 0.279 | 0.210 | 0.221 | 0.431 |

that the features of the held-out set are similar in distribution to the features of the test set, we should expect that choosing a classifier which maximizes the $\ell_2$ margin on the held-out set will lead to good generalization.

Let $f_\theta : \mathcal{X} \to \{-1, 1\}$ be a binary linear classifier defined by

$$f_\theta(x) = x^\top \theta + b.$$

Let $\mathcal{D} := \{(x_i, y_i)\}_{i=1}^n \subseteq \mathcal{X} \times \{-1, 1\}$ be the training set and $\mathcal{C} := \{x_i : y_i f(x_i) > 0\}$ be the set of points correctly classified by $f_\theta$. Then, we define the minimum $\ell_2$-margin of $f_\theta$ to be

$$\min_{x_i \in \mathcal{C}} y_i f_\theta(x_i).$$

The hard margin SVM is a binary linear classifier formulated through the following optimization problem:

$$\min_{\theta, b} \frac{1}{2} ||\theta||_2^2 \text{ s.t. } \forall (x_i, y_i) \in \mathcal{D}, \ y_i(x_i^\top \theta + b) \geq 1.$$

The classifier $\theta_{\text{SVM}}$ learned by a hard margin SVM is the separating hyperplane which maximizes the minimum $\ell_2$-margin. Soudry et al. (2018) show that linear classifiers trained via GD with the unregularized logistic loss converge in direction to $\theta_{\text{SVM}}$. In particular, $||\frac{\theta_{\text{GD}}}{||\theta_{\text{GD}}||_2} - \frac{\theta_{\text{SVM}}}{||\theta_{\text{SVM}}||_2}||_2 \to 0$ at a worst-case rate of $O(\frac{\log \log t}{\log t})$ where $t$ is the number of gradient steps. This result can also be extended to the nonseparable case, though as one might expect, the implicit bias is more complex (Ji & Telgarsky, 2019). The SVM solution was recently studied in the context of group robustness by Chaudhuri et al. (2023), who showed that under certain conditions on the tail of the data distribution, $\theta_{\text{SVM}}$ converges to the classifier with optimal worst-group accuracy as the number of data points goes to infinity.

If LLR maximizes the minimum training margin on the held-out set and this results in a debiased classifier, then we should see a high correlation between the size of group minimum margin and group test accuracy (*i.e.,* minority group points lie near the decision boundary). However, in Table 2, we find that the minimum training margin for a group is not correlated with the test accuracy for that group.

Moreover, convergence towards $\theta_{\text{SVM}}$ is slow in practice. We perform LLR on the held-out sets of Waterbirds, CelebA and CivilComments via SGD with learning rate 0.001 and unregularized logistic loss, and we plot the directional error with $\theta_{\text{SVM}}$ in Figure 3. In line with the slow $O(\frac{\log \log t}{\log t})$ rate from Soudry et al. (2018), we find that even after $10\times$ the number of standard training steps, the LLR classifier has yet to converge to $\theta_{\text{SVM}}$. As the minimum $\ell_2$-margin is not well-correlated with group test accuracy and convergence to $\theta_{\text{SVM}}$ is slow, we conclude that our initial hypothesis does not sufficiently explain the robustness gains of LLR on an imbalanced held-out set.

## 4 LLR performance is primarily due to group balance in the held-out set

The negative results of Section 3 prompted us to reconsider our initial hypothesis. We now present strong evidence for an alternative hypothesis: that the success of LLR is primarily due to *better group balance* in the

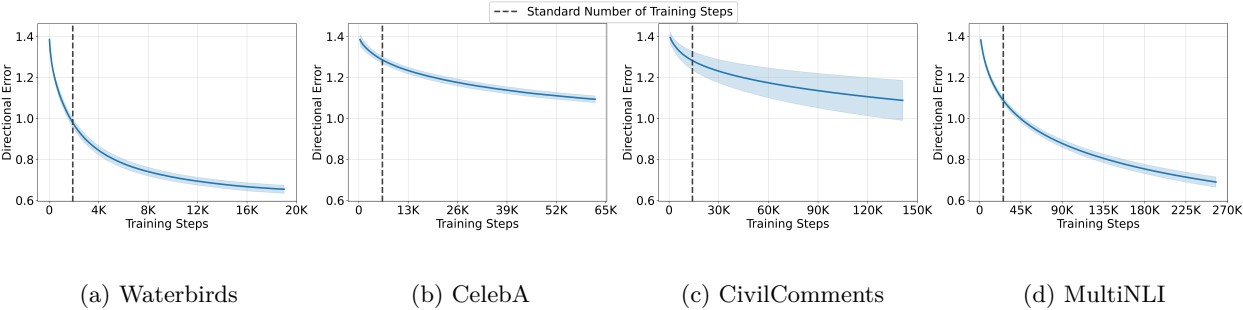

(a) Waterbirds        (b) CelebA        (c) CivilComments        (d) MultiNLI

Figure 3: **Convergence of LLR to the maximum-margin SVM solution is extremely slow.** We plot the mean and standard deviation over 3 experimental seeds of the directional error $\widehat{Err}$ between the last layer weights of a neural network model and an SVM (both trained on the features of the held-out set). We use a ResNet-50 for Waterbirds and CelebA and a BERT-Base for CivilComments and MultiNLI. Here, $\widehat{Err} := ||\frac{\theta_{\mathrm{NN}}}{||\theta_{\mathrm{NN}}||_2} - \frac{\theta_{\mathrm{SVM}}}{||\theta_{\mathrm{SVM}}||_2}||_2$, where $\theta_{\mathrm{NN}}$ denotes the last layer weights and $\theta_{\mathrm{SVM}}$ denotes the weights of an SVM trained on the held-out set features.

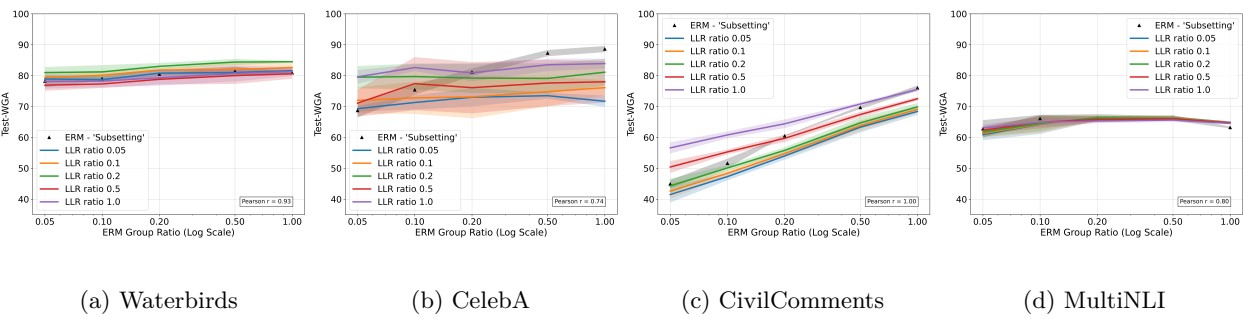

(a) Waterbirds        (b) CelebA        (c) CivilComments        (d) MultiNLI

Figure 4: **LLR performance is determined by held-out set group balance.** We compare the test WGA of ERM and LLR models while controlling the group balance of the training and held-out sets. We use ResNet-50 for the vision datasets and BERT-Base for the language datasets, and we plot the mean and standard deviation over 3 experimental seeds. We compute the Pearson correlation coefficient between the ERM test WGA and the LLR test WGA for each dataset; the presented coefficients are averaged over all 5 group ratios. We find that LLR worst-group accuracy correlates strongly with change in group balance; in particular, LLR tends to improve over ERM if and only if the held-out set has better group balance. Interestingly, we observe a decrease is WGA on MultiNLI as we approach 1:1 group balance. This can be explained by the fact that *optimal* group balance for MultiNLI requires over-representation of the majority groups (Qiao et al., 2025).

held-out set as compared to the training set. In particular, we argue that the "free lunch" interpretation of CB-LLR (LaBonte et al., 2023) is more likely to arise due to improving class balancing over the ERM baseline than an implicit-bias advantage of LLR. Similarly, AFR (Qiu et al., 2023) performs implicit group-balancing via loss reweighting on the held-out set. While LLR may not be as "unreasonably effective" as previously claimed, we do see that it can surprisingly recover the WGA of an optimally class-balanced model even when ERM was not optimally class-balanced. Generally, LLR remains a highly effective technique when group annotations (or proxies thereof) are available only on the held-out set.

## 4.1 LLR worst-group accuracy correlates strongly with ERM vs LLR group ratio

To isolate the effect of group balance on LLR, we perform an ablation study on the ERM and LLR group ratios. We directly control the group ratio — defined as the ratio between the number of minority group points and the number of majority group points — by removing data from the held-out set until the desired group ratio is achieved for each class, keeping the total data in each stage constant. Fixing majority and

minority group sizes also ensures that each dataset is ultimately class-balanced (*i.e.,* we use the *subsetting* class-balancing technique).

We vary the group ratios for both the ERM training set and the LLR held-out set between 0.05 and 1.0. A group ratio of 1.0 corresponds to the majority and minority groups having equal size, whereas a group ratio of 0.05 implies that the minority groups are 5% the size of the majority groups. Our results are detailed in Figure 4. For a fair comparison, the ERM-trained models are trained with the held-out set added in; as an example, on Waterbirds we compare LLR with 20% of data following an ERM with 80% of data (colored line) to an ERM with 100% of data (gray line). Note that LLR with a group ratio of 1.0 corresponds to DFR, as this reflects perfectly group-balanced data.

We find that LLR performance correlates strongly with the change in group balance between the training and held-out datasets. LLR only shows significant gains over ERM when it is trained on a held-out set with a *higher group ratio* than the ERM training set, and vice versa. Moreover, the test WGA of LLR trained with a fixed group ratio is remarkably similar to the test WGA of ERM trained with that same group ratio (observed most impressively on CivilComments). In Figure 4, we detail the average Pearson correlation coefficients between ERM test WGA and LLR test WGA. Specifically, we compute the Pearson correlation coefficients between ERM test WGA (gray line) and LLR test WGA (colored line) for all group ratios in $(0.05, 0.1, 0.2, 0.5, 1.0)$, and average over group ratios. These average correlation coefficients reveal a strong trend, with $r > 0.9$ on Waterbirds and CivilComments. We include the full (non-averaged) coefficients in Appendix A.

These results indicate that the two-stage training procedure of LLR methods is *not* fundamentally oriented towards learning a debiased classifier as we initially hypothesized. Rather, some form of group-balancing is generally necessary to maximally capture LLR robustness benefits. Moreover, our results suggest that LLR methods are limited by how well they improve the group ratio, and thus DFR likely upper bounds the WGA of any LLR method which does not utilize group annotations.

### 4.2 LLR can recover optimally class-balanced WGA

In the previous section, we explicitly controlled group ratios to evaluate LLR worst-group accuracy in a fine-grained ablation study. A remaining question is whether our insights hold in more realistic settings, *i.e.,* across benchmark datasets with standard group ratios and data balancing methodologies. This section answers this question in the affirmative through an empirical study on the interaction between LLR and *class-balancing*, a common data pre-processing method shown to approach more complicated group robustness algorithms (Idrissi et al., 2022). Because groups are defined as a refinement of classes, each of these class-balancing strategies implicitly induces some degree of group-balancing without requiring group labels. This makes class-balancing a natural baseline for understanding when explicit group-balancing is necessary and how much robustness can be achieved without group annotations.

We utilize three standard techniques for class-balancing, detailed in Section 2: *subsetting*, wherein the size of each class is reduced to match that of the smallest class; *upsampling*, wherein sampling probabilities for each class are adjusted so that mini-batches are class-balanced in expectation; and *upweighting*, wherein minority class samples are assigned larger loss weights (LaBonte et al., 2024). We compare ERM and LLR trained with all permutations of these class-balancing methods and display the results in Figure 5.

Our first finding is that our insights from Section 4.1 hold in realistic benchmark datasets with standard group ratios (without our explicit control from earlier). In particular, consider only ERM and LLR with no class-balancing — the leftmost gray and blue "No CB" bars respectively in each subplot of Figure 5. We can see that in all four subplots, No CB LLR either worsens No CB ERM worst-group accuracy, or they lie within one standard deviation of each other. This suggests that without additional intervention in the form of class-balancing, LLR does not improve over ERM with the same group ratio. This is especially apparent in MultiNLI, the only dataset which is class-balanced *a priori*; ERM is more robust than any instantiation of LLR on this dataset, echoing earlier findings of LaBonte et al. (2023).

Our second, and more surprising, finding is that CB-LLR (with any class-balancing method) can recover optimally class-balanced WGA *even when ERM was not optimally class-balanced.* For example, upweighting

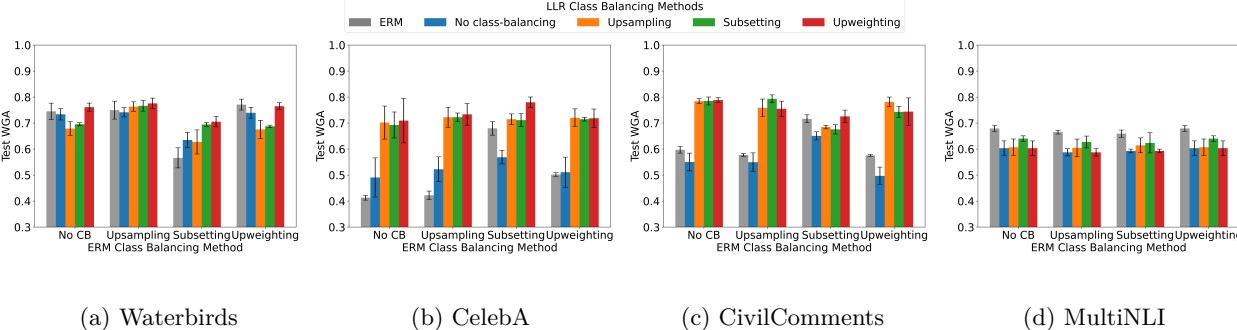

(a) Waterbirds        (b) CelebA        (c) CivilComments        (d) MultiNLI

Figure 5: **LLR can recover optimally class-balanced WGA.** We compare the test WGA of ERM (trained on 100% of the data) against LLR (trained on a held-out subset with the same group ratio). For both approaches, we evaluate four class-balancing strategies: *no class balancing (No CB)*, *upsampling*, *subsetting*, and *upweighting* (defined in Section 2). We use ResNet-50 for the vision datasets and BERT-Base for the language datasets. We plot mean and standard deviation across 3 seeds. Each $x$-axis group corresponds to the class-balancing method used to train the ERM baseline (gray bar). Within each group, we compare this ERM model to the four LLR variants (colored bars), each using the corresponding class-balancing method.

ERM is very poor on CelebA, but running CB-LLR recovers the (best) subsetting ERM worst-group accuracy within one standard deviation. Prior work (LaBonte et al., 2024) has shown that subsetting on Waterbirds, and upsampling and upweighting on CelebA and CivilComments, can cause substantial degradation in WGA. Importantly, this degradation arises from the optimization dynamics of ERM rather than the class distribution itself. Because CB-LLR performs only linear retraining on a fixed representation, it does not suffer from the same instability.

Before moving on, we remark that Figure 5 suggests LLR may yet trump optimally class-balanced ERM in certain edge cases. Specifically: subsetting ERM/upweighting LLR on CelebA, upsampling ERM/subsetting LLR on CivilComments, and upweighting ERM/upsampling LLR on CivilComments exceed all ERMs by at least one standard deviation. It is possible that characteristics of the *specific data selected* for LLR are influential in these cases (see, *e.g.,* previous work by LaBonte et al. (2023); Qiao et al. (2025)).

### 4.3 Recent LLR methods succeed via implicit group-balancing

In this section, we reconcile our findings with the literature by asking the following question: if LLR on an equally-imbalanced held-out set does not significantly improve WGA, how do LLR methods like CB-LLR (LaBonte et al., 2023) and AFR (Qiu et al., 2023) achieve state-of-the-art group robustness without group annotations? We argue that these techniques enjoy success primarily due to *implicit group-balancing*, wherein the held-out set is effectively more group-balanced than the training set due to user interventions.

The CB-LLR results of LaBonte et al. (2023) use class-balanced *upsampling* for both ERM and LLR. Since upsampling is a suboptimal procedure for ERM on CelebA and CivilComments (see Figure 5 as well as LaBonte et al. (2024)), LLR has considerably more room to improve over this baseline. In other words, the strong CB-LLR performance is largely explained by the fact that the ERM model it starts from is itself degraded by suboptimal class-balancing. The improved WGA on Waterbirds is more simply explained: following standard practice (Kirichenko et al., 2023), CB-LLR is run on 50% *of the Waterbirds validation set* (which is more group-balanced!) instead of 20% *of the training set, which is group-imbalanced* as we do in Figure 5.

Compared to standard LLR, which utilizes the cross-entropy loss only, AFR incorporates a weighted loss function which prioritizes points upon which the ERM model performs poorly (detailed in Equation 1). Specifically, AFR introduces a weight for each held-out example $i$ proportional to $\exp(-\gamma \hat{p}_i)$, where $\hat{p}_i$ is the probability for the correct class $y_i$ and $\gamma \geq 0$ is a tunable temperature parameter.[6] Therefore, the AFR held-out set effectively has better group balance than the training set, in the sense of the *upweighting* method. Notably, the ablations of Qiu et al. (2023) show that setting $\gamma = 0$ reduces to CB-LLR with upweighting!

---

[6]Loss-based adjustments are common among group robustness methods not using group annotations, *e.g.,* Liu et al. (2021a).

A positive takeaway from our analysis in this section is that LLR is exceptional at *leveraging data interventions on exclusively the held-out set.* That is, given a principled technique to increase group balance (*e.g.,* class-balancing in CB-LLR, loss-based upweighting in AFR, or disagreement in SELF (LaBonte et al., 2023)), one need only apply the intervention to a limited number of held-out data and retrain the last layer to observe robustness gains matching or exceeding that obtained by applying the same method to ERM (which would require applying the intervention to the much larger training set). Overall, the strength of LLR is its group-annotation and computational efficiency, and it remains a powerful method for group robustness.

## 5  Discussion

In this paper, we proposed and eventually rejected a hypothesis explaining the success of LLR through neural collapse and the implicit bias of gradient descent. However, we also presented strong evidence supporting an alternate hypothesis: that the improved WGA of LLR is primarily due to improved group balance in the held-out set. One consequence of this observation is that LLR can recover the WGA of an optimally class-balanced model, even if ERM was not optimally class-balanced. Overall, LLR remains an effective method to improve worst-group accuracy, especially if a limited number of group annotations are available (*e.g.,* DFR), or in conjunction with implicit group-balancing techniques (*e.g.,* CB-LLR, SELF, and AFR).

Our work highlights the need for further research towards two open questions. First, how do neural networks learn both core and spurious features during ERM training? Recent progress towards this question, *e.g.,* by Qiu et al. (2024), is heartening, but the fact remains that we have no substantial theoretical justification for this key assumption underlying LLR (Izmailov et al., 2022). Second, what is the mechanism by which LLR with better group ratio than ERM reweights the core and spurious features? Addressing this second question is necessary for a complete understanding of LLR, and it is possible this mechanism could be leveraged to design algorithms which target WGA more explicitly than reweighting via logistic regression.

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

# A   Additional experiments

In this section, we present additional experiments deferred from the main body of the work.

## A.1   Explicit group-balancing

First, we study an extension of the implicit group-balancing discussion from Section 4.3 to explicit group-balancing. As discussed in Section 2, we investigate three explicit balancing methods: subsetting, upsampling, and upweighting. It has previously been observed that using upsampling or upweighting for class-balancing can result in drastic decreases to test WGA during training (LaBonte et al., 2024). If the performance improvement from LLR is indeed primarily due to the implicit group-balancing achieved by class-balancing, we would expect similar degradation when we explicitly group-balance using these same techniques.

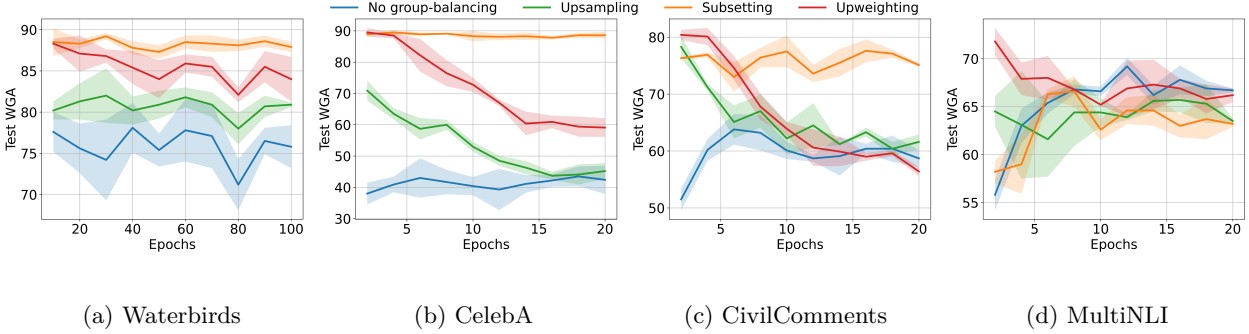

| (a) Waterbirds | (b) CelebA | (c) CivilComments | (d) MultiNLI |

Figure 6: **Group-balancing with upsampling and upweighting can lead to catastrophic collapse.** We compare three group-balancing methods: *subsetting*, *upsampling*, and *upweighting*. We plot the mean and standard deviation over 3 experimental seeds for a ResNet-50 on the vision datasets and a BERT-Base on the language datasets. We note a dramatic decrease in test WGA during training for both upsampling and upweighting on CelebA and CivilComments. We also see a collapse in WGA for upweighting on MultiNLI.

Figure 6 shows how different group-balancing methods affect WGA during training. Similar to the class-balancing results, we find that upsampling and upweighting cause substantial degradation on CelebA and CivilComments. Additionally, group-balanced upweighting leads to degradation on MultiNLI and a mild decrease in WGA on Waterbirds. Overall, the training dynamics induced by subsetting, upweighting, and upsampling for group-balancing resemble those observed for class-balancing in LaBonte et al. (2024): most group-balancing methods behave similarly and do not exhibit degradation.

The key exceptions are MultiNLI—where the dataset is naturally class-balanced but not group-balanced—and subsetting on Waterbirds. For MultiNLI, the lack of class imbalance causes group-balanced upweighting to overweight noisy minority signals, leading to degradation. For Waterbirds, class-balanced subsetting disproportionately removes examples from its already small minority groups, harming performance, whereas group-balanced subsetting avoids this issue and therefore performs well.

In Figure 7, we compare the test WGA of class-balanced ERM and group-balanced DFR. This comparison is motivated by the practical reality that class-balancing is widely used when group annotations are unavailable, and is often assumed to provide a good surrogate for group-balancing. However, recent work suggests that class-balancing can interact with robustness in subtle and dataset-dependent ways. By directly contrasting class-balanced ERM with group-balanced DFR, we aim to isolate the extent to which robustness gains arise from improved representations versus improved group balance. Our results show that while class-balancing affects ERM and DFR substantially, explicit group-balancing still yields consistent WGA improvements, demonstrating its unique value when group annotations—even limited ones—are available.

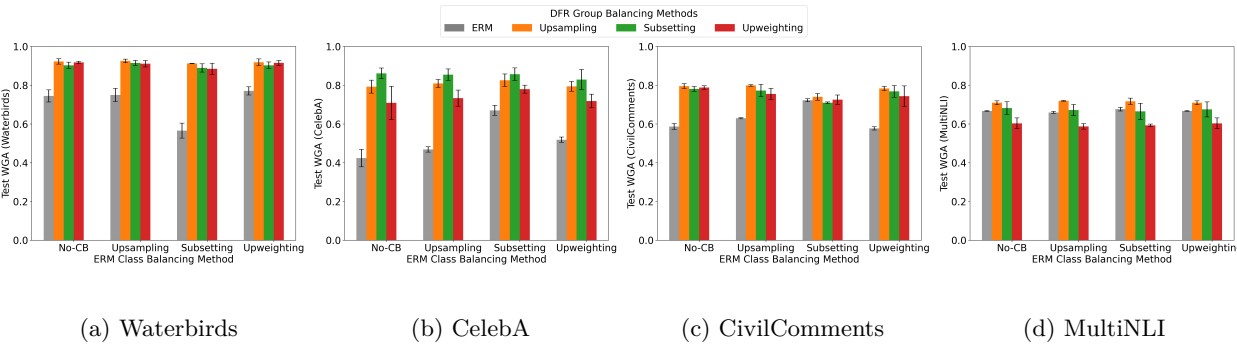

(a) Waterbirds      (b) CelebA      (c) CivilComments      (d) MultiNLI

Figure 7: **Choice of group-balancing method is important to DFR success.** We compare class-balanced ERM to group-balanced DFR across three balancing methods: subsetting, upsampling, and upweighting. We plot the mean and standard deviation over 3 experimental seeds for a ResNet-50 on the vision datasets and a BERT-Base on the language datasets. We note that the choice of balancing method has a dramatic effect on the test WGA for both ERM and DFR. For instance, on CelebA, DFR with subsetting consistently outperforms DFR with upsampling or upweighting, regardless of the ERM balancing method used.

## A.2 Supplemental plots for average accuracy

In this section, we provide corresponding figures to those in the main text displaying average accuracy (AA) instead of worst-group accuracy. While group robustness methods are known in some cases to degrade average accuracy (Sagawa et al., 2020b; Liu et al., 2021a), our results show that LLR generally enjoys similar average accuracy compared to ERM.

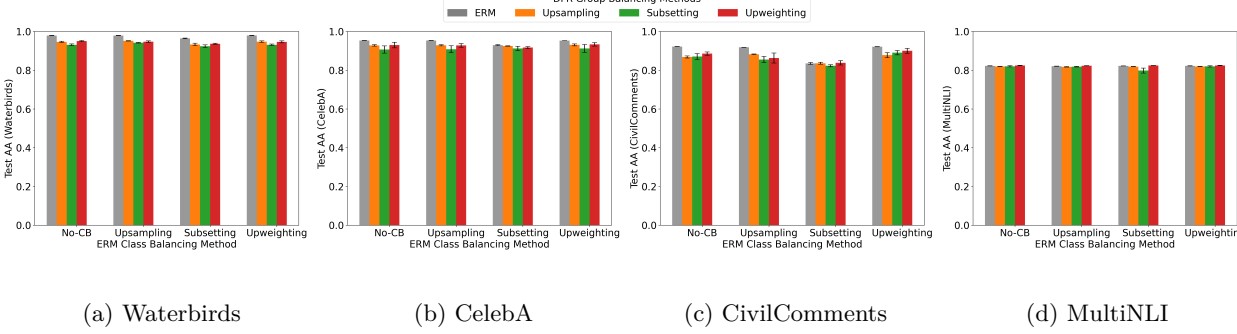

|           |           |                  |             |
| :-------: | :-------: | :--------------: | :---------: |
| (a) Waterbirds | (b) CelebA | (c) CivilComments | (d) MultiNLI |

Figure 8: Test average accuracy (AA) for ERM, upsampling, subsetting, and upweighting across the Waterbirds, CelebA, CivilComments, and MultiNLI datasets. Across all four benchmarks, average accuracy remains uniformly high and nearly identical across class-balancing methods, indicating that these interventions primarily affect minority-group robustness rather than overall predictive performance.

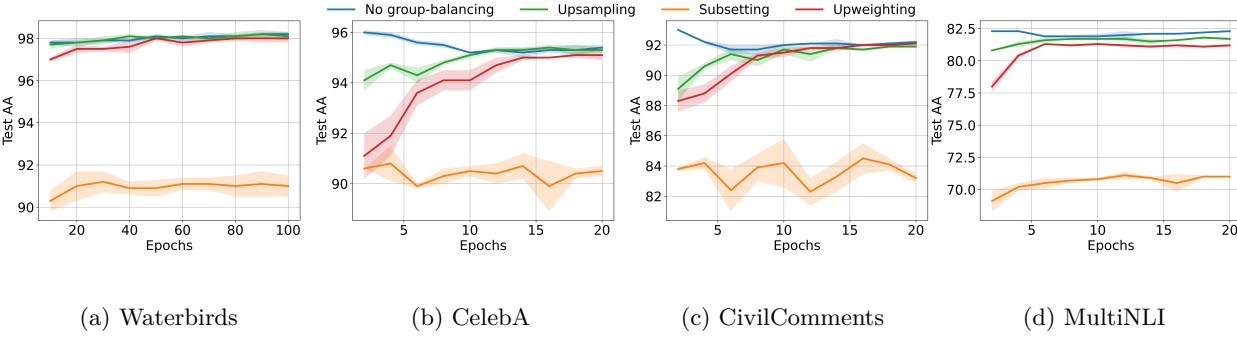

|           |           |                  |             |
| :-------: | :-------: | :--------------: | :---------: |
| (a) Waterbirds | (b) CelebA | (c) CivilComments | (d) MultiNLI |

Figure 9: Test average accuracy (AA) over training epochs for ERM, upsampling, subsetting, and upweighting on the Waterbirds, CelebA, CivilComments, and MultiNLI datasets. With the exception of subsetting — which consistently underperforms due to lack of data – the class-balancing strategies produce only modest variation in average accuracy compared to their much larger impact on worst-group accuracy.

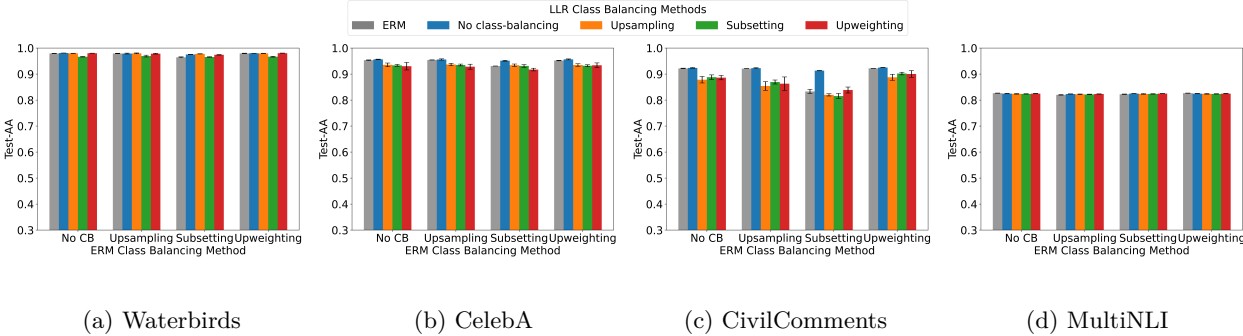

(a) Waterbirds        (b) CelebA        (c) CivilComments        (d) MultiNLI

Figure 10: Test average accuracy of last-layer retraining (LLR) when the underlying ERM model is trained with no class-balancing, upsampling, subsetting, or upweighting, across the Waterbirds, CelebA, CivilComments, and MultiNLI datasets. Across all benchmarks, LLR produces similar Test AA regardless of the ERM class-balancing strategy, indicating that the overall predictive performance of LLR is largely insensitive to the class distribution used during ERM training, even though worst-group accuracy differs substantially in the corresponding WGA results.

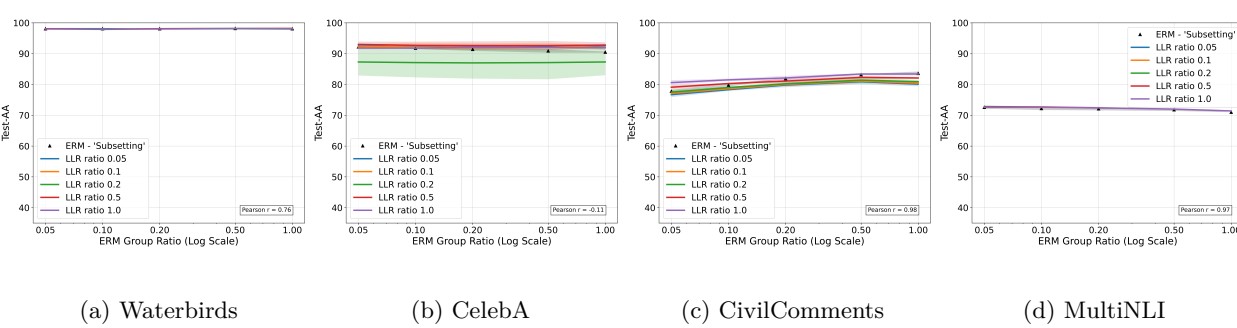

(a) Waterbirds        (b) CelebA        (c) CivilComments        (d) MultiNLI

Figure 11: Test average accuracy of last-layer retraining (LLR) under varying group ratios, evaluated across Waterbirds, CelebA, CivilComments, and MultiNLI. For each dataset, we report LLR performance when applied to ERM models trained with subsetting, upsampling, and no class-balancing. Across all group ratios, Test AA remains remarkably stable, indicating that LLR's overall predictive performance is largely insensitive to the choice of regularization — even though worst-group accuracy can vary substantially under the same settings.

### A.3 Supplemental correlation coefficients

In this section, we provide full Pearson correlation coefficients for the plots with averaged coefficients over group ratios, namely Figure 4 and Figure 11.

Table 3: Full Pearson correlation coefficients for Figure 4. Note that the displayed Pearson correlation coefficients in Figure 4 are the coefficients in this table averaged across rows.

| LLR Group Ratio | 0.05 | 0.1 | 0.2 | 0.5 | 1.0 |
|---|---|---|---|---|---|
| Waterbirds | 0.915 | 0.950 | 0.961 | 0.937 | 0.898 |
| CelebA | 0.793 | 0.929 | 0.348 | 0.820 | 0.834 |
| CivilComments | 0.999 | 0.999 | 0.998 | 0.996 | 0.995 |
| MultiNLI | 0.740 | 0.757 | 0.786 | 0.823 | 0.871 |

Table 4: Full Pearson correlation coefficients for Figure 11. Note that the displayed Pearson correlation coefficients in Figure 11 are the coefficients in this table averaged across rows.

| LLR Group Ratio | 0.05 | 0.1 | 0.2 | 0.5 | 1.00 |
|---|---|---|---|---|---|
| Waterbirds | 0.791 | 0.791 | 0.791 | 0.535 | 0.875 |
| CelebA | -0.707 | 0.195 | -0.022 | 0.457 | -0.468 |
| CivilComments | 0.970 | 0.975 | 0.984 | 0.989 | 0.979 |
| MultiNLI | 0.976 | 0.976 | 0.976 | 0.966 | 0.967 |

### A.4 Swin Transformer Experiments

We reproduce the results from Sections 3 and 4 on Waterbirds and CelebA using a Swin Transformer (Liu et al., 2021b) model instead of a ResNet. For all experiments in this section a Swin-Tiny was used, which has a similar number of parameters as a ResNet-50. We see that the Swin results match those produced by the ResNet, implying that the phenomena discussed in Sections 3 and 4 are agnostic to convolutional vs. Transformer-based models.

Table 5: **Algorithm 1 is drastically more memory efficient for computing $\mathcal{NC}_1$ on Swin-Tiny.** We display the vectorized feature dimensions $N$ for Swin-Tiny on the vision datasets Waterbirds and CelebA. Exact computation of $\mathcal{NC}_1$ requires storing two $N \times N$ matrices of double-precision floating-point numbers, while our estimate of $\mathcal{NC}_1$ computed with Algorithm 1 requires storing only three $N$ dimensional vectors of double-precision floating-point numbers. We see that our stochastic estimate of $\mathcal{NC}_1$ requires many orders of magnitude less memory.

| Dataset | Waterbirds & CelebA |
|---|---|
| Vectorized Feature Dimension $N$ | 37,632 |
| Memory Requirement of Exact $\mathcal{NC}_1$ Computation | 21.10 GiB |
| Memory Requirement of Algorithm 1 | 0.2871 MiB |

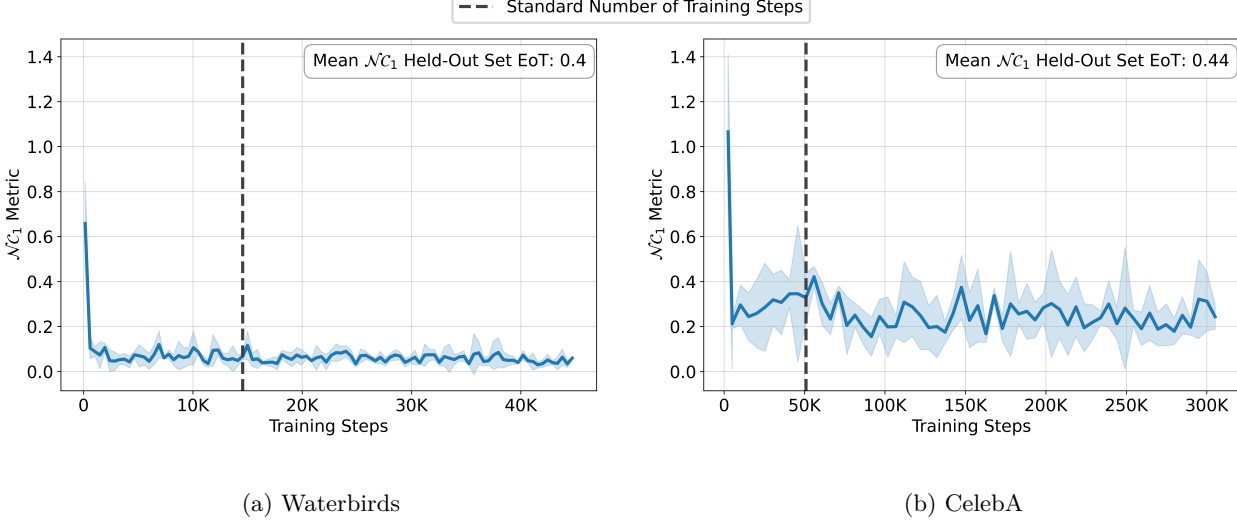

(a) Waterbirds

(b) CelebA

Figure 12: **Collapse of class feature variability in Swin-Tiny occurs after standard ERM training on Waterbirds and CelebA, if at all.** We plot a stochastic estimate of the empirical metric of neural collapse $\mathcal{NC}_1$ using Algorithm 1 throughout training a Swin-Tiny on Waterbirds and CelebA. $\mathcal{NC}_1$ is computed using $K = 10$ random vectors. The mean and standard deviation for $\mathcal{NC}_1$ computed across 3 experimental seeds is displayed. We also display the mean $\mathcal{NC}_1$ metric computed on the features of the held-out set at the end of training (EoT).

Table 6: **Group test accuracy of Swin-Tiny is not well-correlated with the minimum $\ell_2$ margin on the held-out set of Waterbirds and CelebA.** We compute the Pearson correlation coefficient between the individual group test accuracy of a Swin-Tiny and the minimum margin of each group in the held-out set for Waterbirds and CelebA across 3 experimental seeds. We see that held-out set minimum $\ell_2$ margin is not well correlated with test accuracy. A complete breakdown of Waterbirds and CelebA including distribution and group sizes is included in Table 7.

| Dataset | Group 0 | Group 1 | Group 2 | Group 3 |
|---|---|---|---|---|
| Waterbirds | 0.160 | 0.215 | 0.175 | 0.234 |
| CelebA | 0.296 | 0.313 | 0.354 | 0.468 |

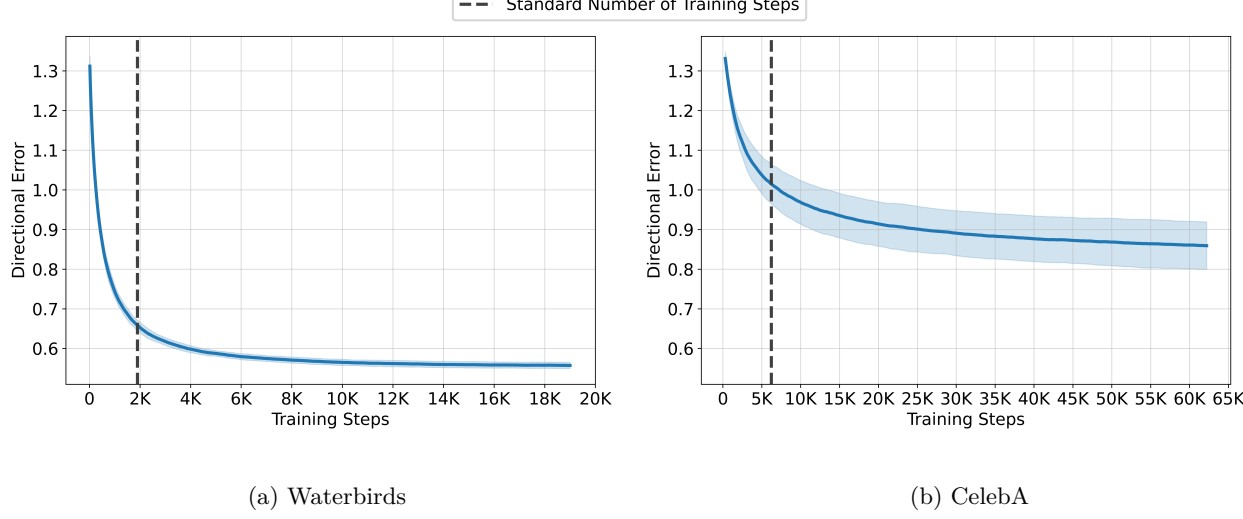

(a) Waterbirds  (b) CelebA

Figure 13: **Convergence of Swin-Tiny last layer to the maximum-margin SVM solution is extremely slow during LLR.** We plot the mean and standard deviation over 3 experimental seeds of the directional error $\widehat{Err}$ between the last layer weights of a neural network model and an SVM (both trained on the features of the held-out set). For the neural network, we use a Swin-Tiny trained on Waterbirds and CelebA. Here, $\widehat{Err} := ||\frac{\theta_{\mathrm{NN}}}{||\theta_{\mathrm{NN}}||_2} - \frac{\theta_{\mathrm{SVM}}}{||\theta_{\mathrm{SVM}}||_2}||_2$, where $\theta_{\mathrm{NN}}$ denotes the last layer weights and $\theta_{\mathrm{SVM}}$ denotes the weights of an SVM trained on the held-out set features.

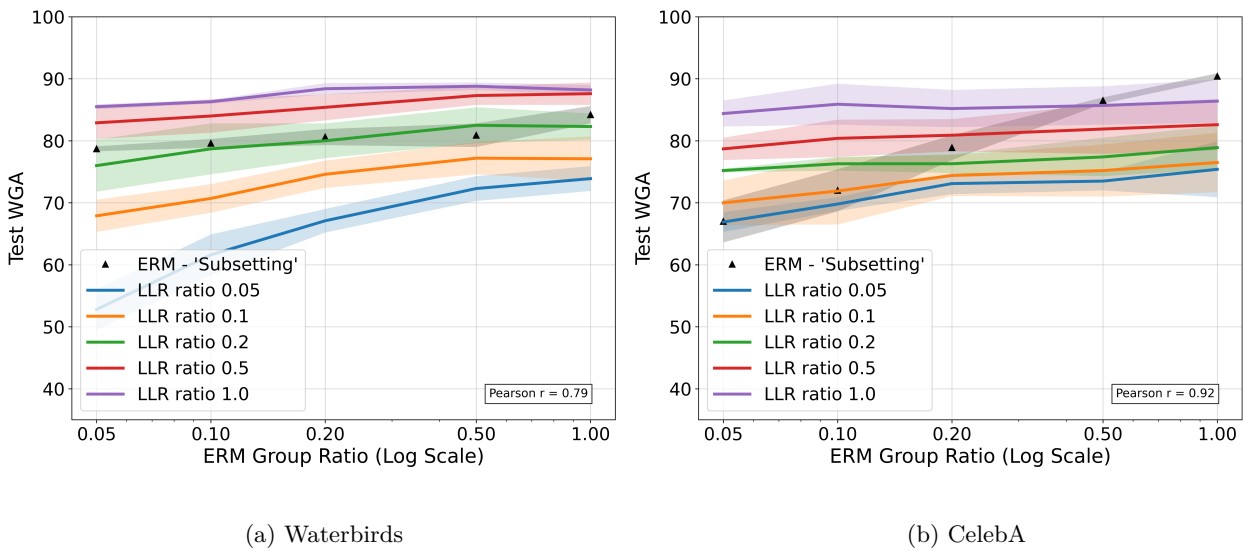

(a) Waterbirds

(b) CelebA

Figure 14: **Swin-Tiny LLR performance is determined by held-out set group balance.** We compare the test WGA of ERM and LLR models while controlling the group balance of the training and held-out sets. In contrast to Figure 4 we use Swin-Tiny for the vision datasets, Waterbirds and CelebA. We plot the mean and standard deviation over 3 experimental seeds. We compute the Pearson correlation coefficient between the ERM test WGA and the LLR test WGA for each dataset; the presented coefficients are averaged over all 5 group ratios. We find that LLR worst-group accuracy correlates strongly with change in group balance; in particular, LLR tends to improve over ERM if and only if the held-out set has better group balance.

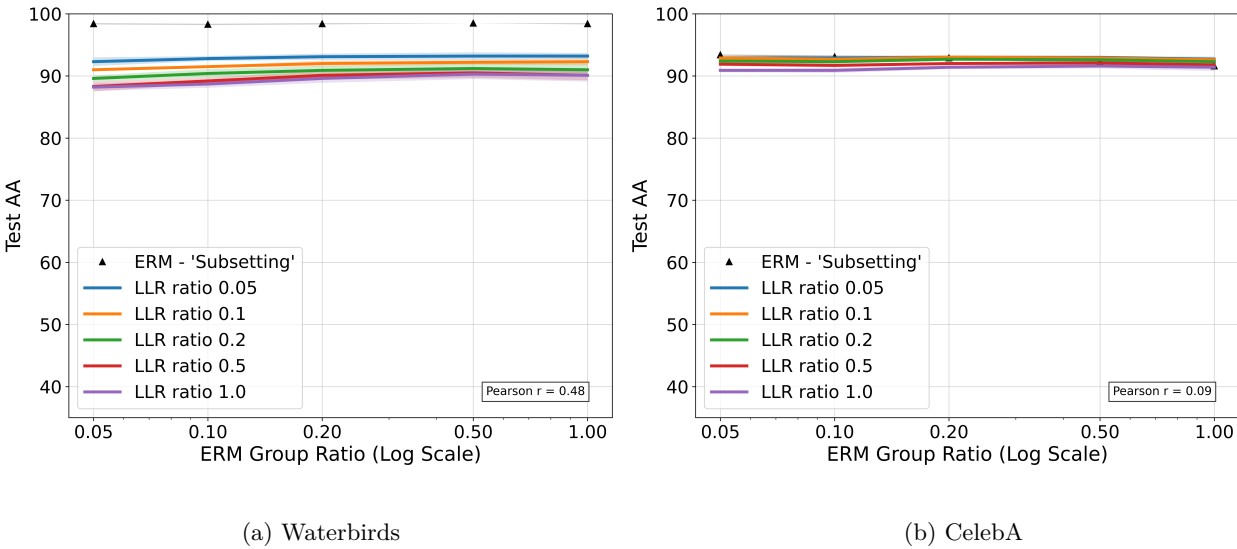

(a) Waterbirds

(b) CelebA

Figure 15: Test average accuracy of LLR trained Swin-Tiny models under varying group ratios, evaluated across Waterbirds and CelebA. For each dataset, we report LLR performance when applied to ERM models trained with subsetting, upsampling, and no class-balancing. Across all group ratios, similar to Figure 11, Test AA remains remarkably stable, indicating that LLR's overall predictive performance is largely insensitive to the choice of regularization — even though worst-group accuracy can vary substantially under the same settings.

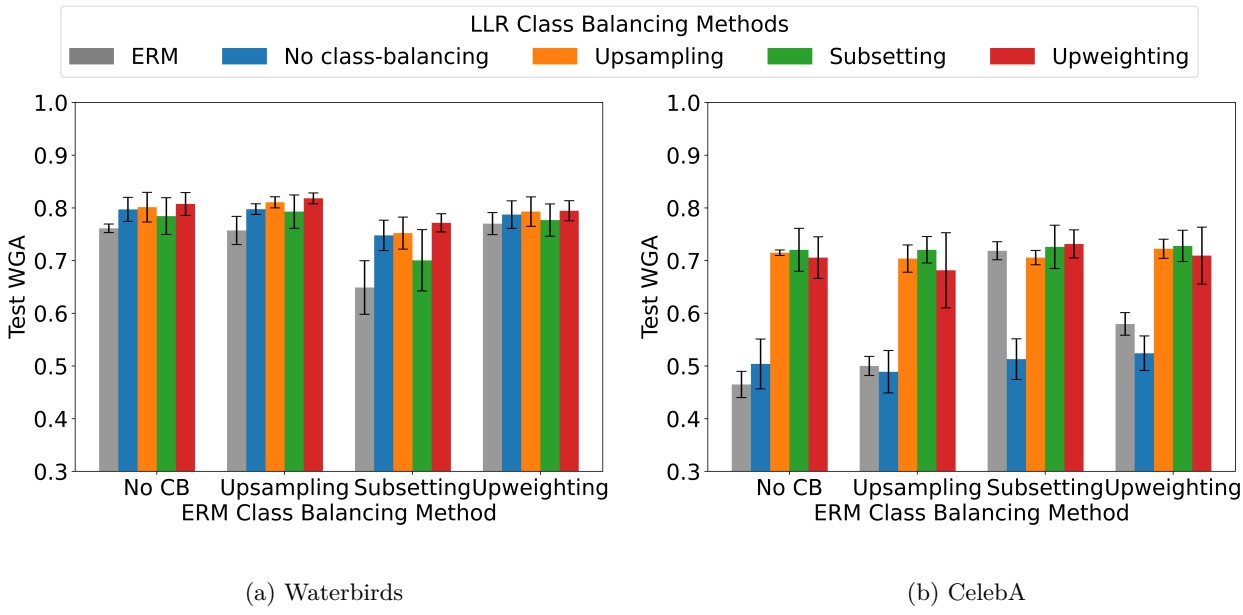

(a) Waterbirds

(b) CelebA

Figure 16: **Swin-Tiny trained with LLR can recover optimally class-balanced WGA.** We compare the test WGA of Swin-Tiny (on Waterbirds and CelebA) trained using ERM (trained on 100% of the data) against Swin-Tiny trained using LLR (trained on a held-out subset with the same group ratio). For both approaches, we evaluate four class-balancing strategies: *no class balancing (No CB)*, *upsampling*, *subsetting*, and *upweighting* (defined in Section 2). Just like in Figure 5, we plot mean and standard deviation across 3 seeds. Each $x$-axis group corresponds to the class-balancing method used to train the ERM baseline (gray bar). Within each group, we compare this ERM model to the four LLR variants (colored bars), each using the corresponding class-balancing method.

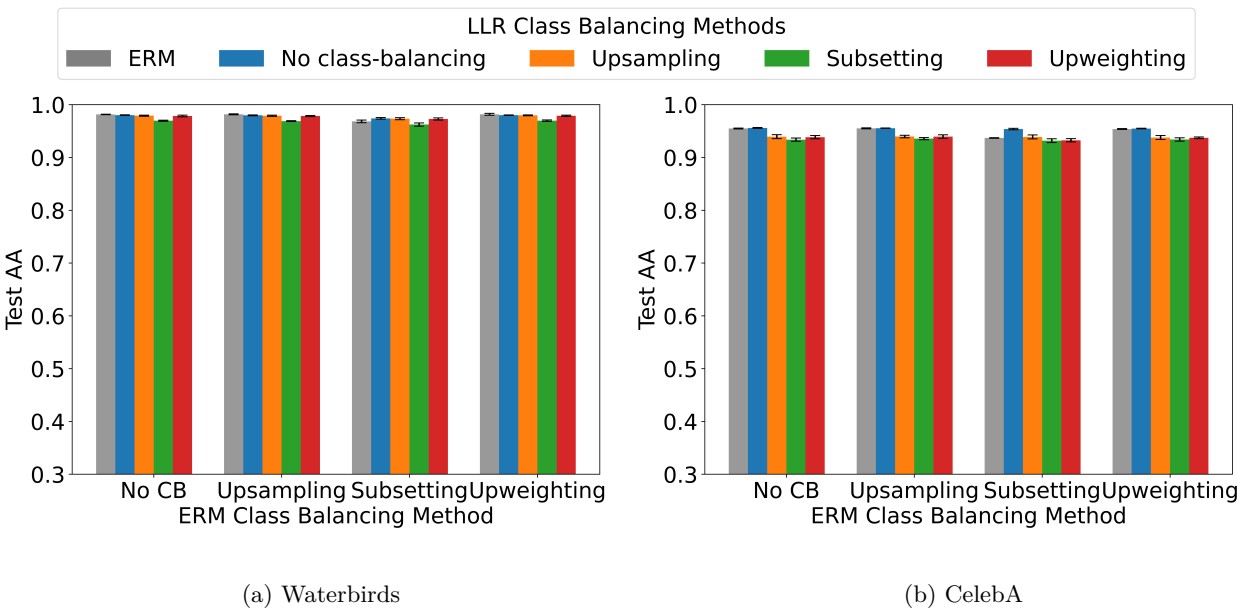

(a) Waterbirds

(b) CelebA

Figure 17: Test average accuracy (AA) for ERM, upsampling, subsetting, and upweighting across the Waterbirds and CelebA datasets. Across both benchmarks, LLR produces similar test AA regardless of the ERM class-balancing strategy, indicating (similar to Figure 10) that the overall predictive performance of LLR is largely insensitive to the class distribution used during ERM training, even though worst-group accuracy differs substantially in the corresponding WGA results.

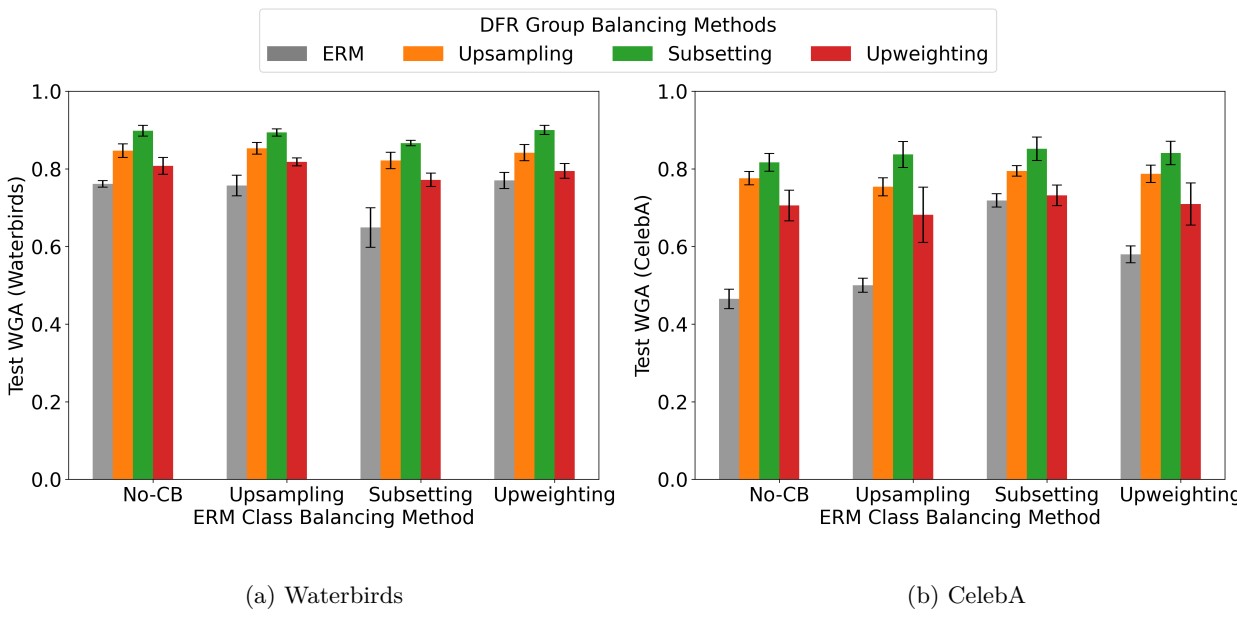

(a) Waterbirds

(b) CelebA

Figure 18: **Choice of group-balancing method is important to Swin-Tiny DFR success.** We compare a Swin-Tiny model trained with class-balanced ERM to a Swin-Tiny model trained with group-balanced DFR across three balancing methods: subsetting, upsampling, and upweighting. We show results for the vision datasets Waterbirds and CelebA. We plot the mean and standard deviation over 3 experimental seeds. We note that, similar to Figure 7 the choice of balancing method has a dramatic effect on the test WGA for both ERM and DFR.

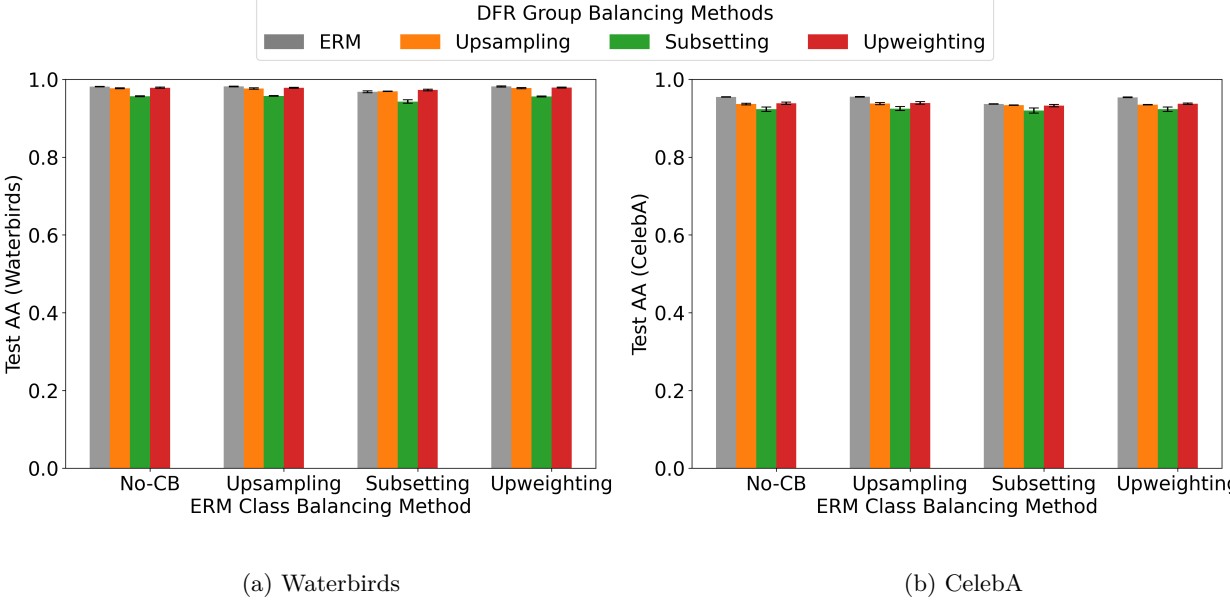

(a) Waterbirds

(b) CelebA

Figure 19: Test average accuracy (AA) for ERM, upsampling, subsetting, and upweighting across the Waterbirds and CelebA datasets. Across both benchmarks, average accuracy remains uniformly high and nearly identical across class-balancing methods, similar to Figure 8. This indicates that these interventions primarily affect minority-group robustness rather than overall predictive performance.

# B    Additional dataset and training details

## B.1    Dataset details

We study four benchmarks for group robustness across vision and language tasks, outlined in Section 2 and detailed in Table 7.

Table 7: **Dataset composition.** The class probabilities change dramatically when conditioned on the spurious feature. Note that Waterbirds is the only dataset that has a distribution shift and MultiNLI is the only dataset which is class-balanced *a priori*. The minority groups within each class are denoted by an asterisk in the "Num" column. Probabilities may not sum to 1 due to rounding.

| Dataset | Group $g$ | | | Training distribution $\hat{p}$ | | | Data quantity | | |
|---|---|---|---|---|---|---|---|---|---|
| | Num | Class $y$ | Spurious $s$ | $\hat{p}(y)$ | $\hat{p}(g)$ | $\hat{p}(y\|s)$ | Train | Val | Test |
| Waterbirds | 0 | landbird | land | .768 | .730 | .984 | 3498 | 467 | 2225 |
| | 1* | landbird | water | | .038 | .148 | 184 | 466 | 2225 |
| | 2* | waterbird | land | .232 | .012 | .016 | 56 | 133 | 642 |
| | 3 | waterbird | water | | .220 | .852 | 1057 | 133 | 642 |
| CelebA | 0 | non-blond | female | .851 | .440 | .758 | 71629 | 8535 | 9767 |
| | 1* | non-blond | male | | .411 | .980 | 66874 | 8276 | 7535 |
| | 2 | blond | female | .149 | .141 | .242 | 22880 | 2874 | 2480 |
| | 3* | blond | male | | .009 | .020 | 1387 | 182 | 180 |
| CivilComments | 0 | neutral | no identity | .887 | .551 | .921 | 148186 | 25159 | 74780 |
| | 1* | neutral | identity | | .336 | .836 | 90337 | 14966 | 43778 |
| | 2* | toxic | no identity | .113 | .047 | .079 | 12731 | 2111 | 6455 |
| | 3 | toxic | identity | | .066 | .164 | 17784 | 2944 | 8769 |
| MultiNLI | 0 | contradiction | no negation | .333 | .279 | .300 | 57498 | 22814 | 34597 |
| | 1* | contradiction | negation | | .054 | .761 | 11158 | 4634 | 6655 |
| | 2 | entailment | no negation | .334 | .327 | .352 | 67376 | 26949 | 40496 |
| | 3* | entailment | negation | | .007 | .104 | 1521 | 613 | 886 |
| | 4 | neither | no negation | .333 | .323 | .348 | 66630 | 26655 | 39930 |
| | 5* | neither | negation | | .010 | .136 | 1992 | 797 | 1148 |

## B.2 Training details

We utilize ResNet-50 (He et al., 2016) and Swin Transformer Tiny (Liu et al., 2021b) pretrained on ImageNet-1K (Russakovsky et al., 2015) for Waterbirds and CelebA. We also utilize a BERT-Base (Devlin et al., 2019) model pretrained on Book Corpus (Zhu et al., 2015) and English Wikipedia for CivilComments and MultiNLI. These pretrained models are used as the initialization for ERM finetuning under the cross-entropy loss. We use standard ImageNet normalization with standard flip and crop data augmentation for the vision tasks and BERT tokenization for the language tasks (Izmailov et al., 2022). Our implementation uses the following packages: `NumPy` (Harris et al., 2020), `PyTorch` (Paszke et al., 2017; 2019), `Lightning` (Falcon & the PyTorch Lightning maintainers and contributors, 2019), `TorchVision` (TorchVision maintainers and contributors, 2016), `Matplotlib` (Hunter, 2007), `Transformers` (Wolf et al., 2020), and `Milkshake` (LaBonte, 2023).

To our knowledge, the licenses of Waterbirds and CelebA are unknown. CivilComments is released under the CC0 license, and information about MultiNLI's license may be found in Williams et al. (2018).

Our experiments were conducted on two local 24GB Nvidia RTX A5000 GPUs. We provide our ERM finetuning hyperparameters in Table 8. Our LLR experiments were run for the same number of epochs as ERM (unless otherwise mentioned) on a held-out dataset (20% of the training set for Waterbirds and half the validation set for the other three datasets). We used SGD with learning rate 0.01 with no regularization or learning rate schedule, except for Figure 3 where we used learning rate 0.001.

Table 8: **Default ERM Hyperparameters.** The below table details our default hyperparameters for *ERM only*. Note that *for LLR only*, we used SGD with learning rate 0.01 with no regularization or learning rate schedule, except for Figure 3 where we used learning rate 0.001.

| Dataset | Optimizer | Initial LR | LR schedule | Batch size | Weight decay | Epochs |
|---|---|---|---|---|---|---|
| Waterbirds | AdamW | $1 \times 10^{-5}$ | Cosine | 32 | $1 \times 10^{-4}$ | 100 |
| CelebA | AdamW | $1 \times 10^{-5}$ | Cosine | 32 | $1 \times 10^{-4}$ | 20 |
| CivilComments | AdamW | $1 \times 10^{-5}$ | Linear | 32 | $1 \times 10^{-4}$ | 20 |
| MultiNLI | AdamW | $1 \times 10^{-5}$ | Linear | 32 | $1 \times 10^{-4}$ | 20 |

## C $\widehat{\mathcal{NC}}_1$ **Estimator Variance**

As part of our analysis of Algorithm 1 we provide a description of the variance of the estimate $\widehat{\mathcal{NC}}_1$. Importantly, Proposition 1 enables us to compute the relative variance, defined as $Var(\widehat{\mathcal{NC}}_1)/(\mathbb{E}[\widehat{\mathcal{NC}}_1])^2$.

**Proposition 1.** *Let $M = \Sigma_A \Sigma_R^\dagger \in \mathbb{R}^{N \times N}$ be the matrix whose trace is estimated in Algorithm 1. Let $\{z_j\}_{j=1}^K$ be $K$ independent random vectors sampled from a standard multivariate normal distribution $\mathcal{N}(0, I_N)$. The estimator $\widehat{\mathcal{NC}}_1$ is defined as:*

$$\widehat{\mathcal{NC}}_1 = \frac{1}{K|\mathcal{Y}|} \sum_{j=1}^K z_j^\top M z_j. \tag{2}$$

*The variance of this estimator is given by:*

$$Var(\widehat{\mathcal{NC}}_1) = \frac{1}{K|\mathcal{Y}|^2} \left[ tr(MM^\top) + tr(M^2) \right]. \tag{3}$$

*Equivalently, letting $S = \frac{1}{2}(M + M^\top)$ denote the symmetric part of $M$, the variance is:*

$$Var(\widehat{\mathcal{NC}}_1) = \frac{2}{K|\mathcal{Y}|^2} \|S\|_F^2, \tag{4}$$

*where $\|\cdot\|_F$ denotes the Frobenius norm.*

*Proof.* Let $s_j = z_j^\top M z_j$. Since the vectors $z_j$ are independent and identically distributed, the variance of the sample mean is:

$$\text{Var}(\widehat{\mathcal{NC}}_1) = \text{Var}\left( \frac{1}{K|\mathcal{Y}|} \sum_{j=1}^K s_j \right) = \frac{1}{K|\mathcal{Y}|^2} \text{Var}(s_1). \tag{5}$$

We now derive $\text{Var}(s)$ for a single vector $z \sim \mathcal{N}(0, I_N)$. By definition, $\text{Var}(s) = \mathbb{E}[s^2] - (\mathbb{E}[s])^2$.

**First Moment.** Using the linearity of expectation and the fact that $\mathbb{E}[z_i z_k] = \delta_{ik}$ (the Kronecker delta):

$$\mathbb{E}[s] = \mathbb{E}\left[ \sum_{i,k} M_{ik} z_i z_k \right] = \sum_{i,k} M_{ik} \delta_{ik} = \sum_i M_{ii} = \text{tr}(M). \tag{6}$$

**Second Moment.** We expand the square $s^2 = (\sum_{i,k} M_{ik} z_i z_k)(\sum_{p,q} M_{pq} z_p z_q)$ and take the expectation:

$$\mathbb{E}[s^2] = \sum_{i,k,p,q} M_{ik} M_{pq} \mathbb{E}[z_i z_k z_p z_q]. \tag{7}$$

By Isserlis' Theorem (Isserlis, 1918) for Gaussian variables, the fourth moment is:

$$\mathbb{E}[z_i z_k z_p z_q] = \delta_{ik}\delta_{pq} + \delta_{ip}\delta_{kq} + \delta_{iq}\delta_{kp}. \tag{8}$$

Substituting this back into the sum, we obtain three terms:

1. $\sum_{i,k,p,q} M_{ik} M_{pq} \delta_{ik}\delta_{pq} = (\sum_i M_{ii})(\sum_p M_{pp}) = (\text{tr}(M))^2.$

2. $\sum_{i,k,p,q} M_{ik} M_{pq} \delta_{ip}\delta_{kq} = \sum_{i,k} M_{ik} M_{ik} = \text{tr}(MM^\top).$

3. $\sum_{i,k,p,q} M_{ik} M_{pq} \delta_{iq}\delta_{kp} = \sum_{i,k} M_{ik} M_{ki} = \text{tr}(M^2).$

Thus, $\mathbb{E}[s^2] = (\text{tr}(M))^2 + \text{tr}(MM^\top) + \text{tr}(M^2)$.

**Variance Calculation.** The variance for a single sample is:

$$\text{Var}(s) = \mathbb{E}[s^2] - (\mathbb{E}[s])^2 = \text{tr}(MM^\top) + \text{tr}(M^2). \tag{9}$$

Since $z^\top M z = z^\top S z$ for the symmetric part $S = \frac{1}{2}(M + M^\top)$, and for any symmetric matrix $S$, $\text{tr}(SS^\top) = \text{tr}(S^2) = \|S\|_F^2$, we have:

$$\text{Var}(s) = 2\text{tr}(S^2) = 2\|S\|_F^2. \tag{10}$$

Combining this with the scaling factors $K$ and $|\mathcal{Y}|$, we arrive at the final expression:

$$\text{Var}(\widehat{\mathcal{NC}}_1) = \frac{1}{K|\mathcal{Y}|^2}\left[\text{tr}(MM^\top) + \text{tr}(M^2)\right] = \frac{2}{K|\mathcal{Y}|^2}\|S\|_F^2. \tag{11}$$

$\square$

Let us now analyze the relative variance of $\widehat{\mathcal{NC}}_1$. Recall that the relative variance is given by: $\text{RelVar}(\widehat{\mathcal{NC}}_1) := \text{Var}(\widehat{\mathcal{NC}}_1)/(\mathbb{E}[\widehat{\mathcal{NC}}_1])^2$. By Proposition 1, we have

$$\text{RelVar}(\widehat{\mathcal{NC}}_1) = \frac{\text{tr}(MM^\top) + \text{tr}(M^2)}{K(\text{tr}(M))^2}. \tag{12}$$

Let $\{\lambda_i\}_{i=1}^N$ be the eigenvalues of $M$ and let $\{\sigma_i\}_{i=1}^N$ be the singular values of $M$. The expression becomes:

$$\text{RelVar}(\widehat{\mathcal{NC}}_1) = \frac{\sum_i \sigma_i^2 + \sum_i \lambda_i^2}{K(\sum_i \lambda_i)^2}. \tag{13}$$

Under the assumption of a bounded spectral density for both the singular values and eigenvalues, the numerator scales as $O(N)$ while the denominator scales as $O(N^2)$. Consequently, the relative variance scales as $O(1/KN)$. This implies that as the feature dimension $N$ increases, the number of random samples $K$ required to maintain a constant relative error actually decreases, making the Hutchinson estimator increasingly efficient for large-scale models.

## D    Broader impact statement

This work advances equitable and responsible AI development by providing methods to improve robust performance of machine learning models on underrepresented groups, directly mitigating algorithmic bias and promoting fairness across diverse user populations. Our research intends to produce systems that offer more equitable outcomes, thereby increasing public trust and widening the societal benefits of AI technology. However, the responsible deployment of these models, particularly in high-stakes applications, necessitates additional comprehensive, multi-metric evaluation on a full spectrum of fairness criteria to prevent unintended negative consequences and uphold the highest ethical standards.

