# OpenReview forum: "On the Unreasonable Effectiveness of Last-layer Retraining"
_TMLR — Accepted by TMLR_

### Review · Reviewer_rMVs · 2026-01-02

**Summary Of Contributions:**

The paper provides an analysis of last layer retraining to understand how and why it is effective. The paper tries to relate concepts of representation collapse, classifier bias, and spurious features. Using these concepts as the basis for analysis, the authors present evidence of interpretation for why LLR works, that LLR performance comes from better group balance in the held-out dataset.

Strengths:
- The paper is clearly written, and the provided evidence is clearly stated.
- Good understanding on concepts of collapse, classifier bias, and spurious features is provided.
- Trying to relate these three concepts is interesting and novel.
- The approximation algorithm 1 for the neural collapse metric seems quite interesting to me (using hutchinson trace estimator). This seems like a major contribution that can benefit a lot of future studies in the area.

Weaknesses/Questions:
- The paper can improve its structure a little to clarify how each piece of evidence supports/refutes its hypothesis and contributions. It's a little difficult to parse the logical structure since it seems a little bit 'windy' in its current state. For example, it's not immediately obvious that Figure 1 is illustrating a potential issue, and the assumptions on spurious and core features are not necessarily correlated (or not) exactly as illustrated.
- I'm not sure why "better group balance" is a surprising contribution or why its different than the general concept of better distribution alignment. At a minimum, it would help to clearly define what the authors mean by "better group balance". Additionally, this seems just to be a well understood domain adaptation issue where the held-out set used to retrain the last layer is more aligned with the test set (or less aligned to the overfitted data). Could the authors clarify why "group balance" is the appropriate interpretation here that may generalize to LLR as a whole? Or is it a more limited contributions given the way the datasets were processed (to amplify these properties)?
- The other major concern I have is the practical value of such a problem and the real-world impact. I would think that for a sufficiently large dataset, that the spurious correlations across groups (minority vs majority) are going to be similar. Therefore, whether there is "better group balance" or not perhaps would not matter in practice? Also, based on the discussion in “Datasets”, how are spurious associations different for the majority and minority class? Establishing that spurious correlations for minority class is different than the majority class seems to be necessary to refute the initial hypothesis based on my understanding of figure 1b.

**Audience:**

Yes

**Audience Explanation:**

The paper seeks to understand why LLR is beneficial. This is useful for a broad range of ML tasks which can use the findings of the paper to apply its use generally.

**Broader Impact Concerns:**

No concerns.

**Claims And Evidence:**

No

**Claims Explanation:**

I think in general the authors present good empirical evidence for each of their contributions. The approximation algorithm is a good novel contribution. However, there are a few areas where the evidence can be more clearly related.

As written, I'm not completely convinced that "better group balance" is a unique or interesting concept to explain LLR effectiveness or if its just a different name for concepts which can be understood in much simpler terms. I.e., it's an issue of data distribution alignment and the effects seen is no different than what would happen when applying sampling or bagging techniques with the use of a simple classifier. It may help the paper if there is strong evidence the ML community believes LLR functions due to other ways.

**Requested Changes:**

- I wonder if you can include an experiment where the spurious are correlated, inversely correlated, or not correlated and setup these differences in the training and testsets (i.e. correlated training and uncorrelated testing)? There is some examination of this based on the way the datasets are set up, however I think it will be hard to conclude that the group balance is what primarily causes LLR to improve its performance without more thorough analysis here.
- Additional writing improvements to organize evidence and argument to more explicitly relate better group balancing with LLRs. These concepts of spurious correlations, groups, LLR, WGA, reweighting, etc, are quite complex and the relationships between them can be made more obvious.

---

> ### Author Response · Authors · 2026-02-16
> **Author response**
>
> We graciously thank Reviewer rMVs for their insightful comments and questions. We appreciate the reviewer’s comments that our paper is “clearly written” as well as the interaction between neural collapse and classifier bias being “interesting and novel”. We also appreciate that the reviewer recognizes our proposed Hutchinson estimator as a “major contribution that can benefit a lot of future studies” -- we hope this technique encourages additional investigation of neural collapse in modern large-scale settings.
>
> Below, we respond to each of the reviewer’s points.
>
> 1. Regarding writing improvements: thank you for the pointers! It is important to us that our work is accessible to both experts and non-experts. To this end, we have made the following writing changes in the revision: (a) We have adjusted the language throughout regarding neural collapse to achieve more consistency and clarity. (b) We have provided a more detailed explanation of what we mean by “better group balance”. We hope this will make the paper more digestible (in particular, for non-experts). (c) We revised the caption of Figure 1 in the hopes of providing a better visual introduction to our hypothesis. We especially hope that it is more clear now why the initial classifier shown in Figure 1a represents a potential issue.
>
> 2. We appreciate the reviewer’s valuable comments regarding the intricacies of group balance. We define a held-out set to have “better group balance” than an ERM training set when the majority and minority groups in the held-out set are more uniformly represented. In this sense, we believe group balance exactly corresponds to the reviewer’s notion of data distribution misalignment. To be clear, we are not proposing group balance to be some novel metric.
>
>     We would like to reinforce some essential context: as mentioned in the Introduction, our paper is a direct response to the findings of [1, 2], who showed that LLR on a held-out set with the **same data distribution** as the training set can still improve WGA. In this case, it seems impossible that data distribution misalignment could be a factor, which led [1] to term LLR a “free lunch”. Our main contribution in Section 4 is that, despite the framing of [1, 2], we show the robustness improvements of LLR **can still be attributed to group balance**. The surprising part is that the improvement in data distribution alignment is completely implicit: it is induced by the class-balancing methodology of [1] and the loss-based upweighting technique of [2].
>
> 3. With regards to practical value, we believe that spurious correlations are one of the most pervasive fairness issues in machine learning, and have already impacted ML application areas including medicine [3] and criminal justice [4]. Moreover, the literature has shown that larger datasets are **more** susceptible to spurious correlations than smaller ones, due to a greater diversity of classes and features. The Salient ImageNet line of work [5, 6] is highly relevant here: these works use interpretability techniques to discover spurious correlations in ImageNet, which is much larger than the datasets we study. In particular, [6] record over 630 spurious class-feature dependencies, including racial biases.
>
>     We don’t quite understand your comments about the spurious feature being different for majority and minority groups, but we are happy to clarify if you can elaborate further. It may be helpful for us to clarify the definitions of majority and minority groups: Given a label set Y and a spurious feature which takes values in S, the groups are defined by the Cartesian product of Y and S.  Hence, the **spurious feature** is the same for both groups, but its **value** is what changes. For example, in CelebA we have Y = {blond, non-blond} and S = {male, female}; since there are about 6x more blond females than blond males in the dataset, we have the majority group (blond, female) and minority group (blond, male).

---

> ### Author Response · Authors · 2026-02-16
> **Author response**
>
> 4. Thank you for the interesting question regarding control of the spurious correlation. We believe our experiments in Section 4.1 constitute the most comprehensive investigation of your question: they are carefully controlled, as group balance is the only independent variable, and they imply strong correlation between LLR group accuracy and ERM vs LLR group ratio with non-trivial models and benchmark datasets.
>
>     With that said, we have spent some time over the rebuttal period to examine your question in a synthetic data setting with zero confounders. This turned out to be trickier than expected: in such simple settings, neural networks tend to either fit only the spurious feature (making LLR useless, since it is predicated on ERM learning a few core features [7]), or fit only the core feature (making LLR unnecessary).
>
>     One setting we found to give interesting results was the following: Suppose we receive data $x$ drawn from the Boolean hypercube in $d$ dimensions, and let the target be the Boolean XOR function on the first two dimensions, i.e., $y=-x_1 x_2$. Then, let $x_3$ be spuriously correlated with the target as follows: with probability $1-\lambda$ we have $x_3=y$, and with probability $\lambda$ we have $x_3=-y$. This setting is interesting because the target function (a quadratic function in two variables) is strictly more complex than the spurious function (a linear function in one variable); this type of complexity gap has recently been related to the success of LLR by [8].
>
>     We set $d=100$ and train a three-layer neural network with 50 neurons using SGD with learning rate 0.05 and batch size 32 for 1000 steps, with spurious correlation strength $\lambda=0.05$. (That is, the spurious feature matches the label 95% of the time). Then, we perform LLR while varying $\lambda$ for the held-out set, and we report the accuracy on a group-balanced test set as a proxy for WGA. Averaged over 10 random seeds, we find an accuracy of 93.9% with ERM.
>     * For held-out set $\lambda=0.01$, we find an accuracy of 93.8% (-0.01% under ERM).
>     * For held-out set $\lambda=0.02$, we find an accuracy of 93.9% (equal to ERM).
>     * For held-out set $\lambda=0.05$, we find an accuracy of 94.1% (+0.02% over ERM). Note that this corresponds to LLR with the same group balance as ERM.
>     * For held-out set $\lambda=0.20$, we find an accuracy of 95.0% (+1.1% over ERM).
>     * For held-out set $\lambda=0.50$, we find an accuracy of 96.8% (+2.8% over ERM). Note that this corresponds to DFR on a group-balanced held-out set.
>
>     Overall, these results support the conclusions of Section 4.1. In particular, running LLR on a less group-balanced held-out set can harm performance compared to ERM. On the other hand, LLR performance increases monotonically as the held-out set becomes more balanced.
>
> [1] LaBonte et al. “Towards Last-layer Retraining for Group Robustness with Fewer Annotations.” NeurIPS 2023.
>
> [2] Qiu et al. “Simple and Fast Group Robustness by Automatic Feature Reweighting.” ICML 2023.
>
> [3] Zech et al. “Variable generalization performance of a deep learning model to detect pneumonia in chest radiographs: A cross-sectional study.” PLoS Medicine 2018.
>
> [4] Alexandra Chouldechova. “Fair prediction with disparate impact: A study of bias in recidivism prediction instruments.” FATML 2016.
>
> [5] Singla and Feizi. “Salient ImageNet: How to discover spurious features in Deep Learning?” ICLR 2022.
>
> [6] Moayeri et al. “Spuriosity Rankings: Sorting Data to Measure and Mitigate Biases.” NeurIPS 2023.
>
> [7] Kirichenko et al. “Last Layer Re-Training is Sufficient for Robustness to Spurious Correlations.” ICLR 2023.
>
> [8] Qiu et al. “Complexity Matters: Dynamics of Feature Learning in the Presence of Spurious Correlations.” ICML 2024.

---

### Review · Reviewer_souW · 2026-02-02

**Summary Of Contributions:**

This paper studies last-layer retraining (LLR) and how it could improve the worst-group accuracy. It tries to overthrow the common hypothesis of "the success of LLR is from mitigating neural collapse through the held-out set" with expirical evidences. Then, it presents an alternative hypothesis that "the success of LLR is from the better group balance in the held-out set".

There are mainly three parts in this paper. The first two sections is about background and preliminaries; Section 3 is about why the current "mitigating neural collapse" hypothesis does not hold; Section 4 is about the alternative group balance hypothesis and some empirical evidence.

As a side product of this research, a new algorithm of memory efficient computation of NC1 (Neural Collapse 1) estimation is proposed.

I think the overall story is pretty good and for sure interesting, that the common hypothesis is wrong and a new hypothesis is proposed. But I do find a lot of confusions at least for me. Please refer to the following review sections.

**Additional Comments:**

I actually wanted to make an apology for my overdue review. I have the following reasons to help AC understand my background:

(1) I do ML research myself (with first-author top-venue papers published annually), but I haven't read any paper specifically on last-layer retraining. So I need to get familiar with the concepts and terminologies before reviewing this paper, and I spent much time reading the preliminaries and related works sections.

(2) This paper is unusual for me. I review 20+ papers annually, but I haven't read any paper in this style (overthrow common hypothesis and propose a new hypothesis, with empirical evidence). Also, it's my first time reviewing TMLR.

(3) Except for grammar correction, I did not use any AI tools while reviewing this manuscript.

**Audience:**

Yes

**Audience Explanation:**

This paper would be very interesting for the imbalanced machine learning community. And, overthrowing a common hypothesis is always interesting (as well as risky, but that's how science progresses).

**Broader Impact Concerns:**

N/A. I don't think there is any notable ethical concern about this research work.

But we do need to keep in mind that imbalanced machine learning may cover datasets of different demographic groups. When those datasets are used, we always need to keep privacy and follow the license of the dataset.

**Claims And Evidence:**

No

**Claims Explanation:**

For the above question, I wouldn't say a firm no, but at least it's not a firm yes. The authors try to support their claim with empirical evidence. There are many experiments conducted, but I don't feel convinced after reading them.

I try to enumerate my concerns in the order of, from the beginning of the paper to the end. But I'm writing my review after completing reading the paper I cannot guarantee a clear structure of my concerns. Also, please correct me if I made any mistakes/misunderstanding.

(1) The coverage is not enough from my perspective. Only four datasets from two modalities (vision, text) and two ML models (ResNet 50 for vision and BERT-Base) are studied. This means the empirical results may not be generalizable to other data and ML models, especially for pre-trained encoders that are widely fine-tuned nowadays.

(2) In section 3, the authors proposed an estimator to measure neural collapse in a memory-efficient manner. First of all, I don't really understand what's the point of using an estimator. Since we are trying to overthrow a common hypothesis why bother to introduce more uncertainty/estimation? Secondly, even if the estimator is introduced as it is, there must be some theoretical/empirical analysis of how good the estimation is. Table 1 only compares memory reduction but I do believe accuracy matters more than efficiency for the presented research.

(3) By some derivations, neural collapse is formalized by NC1 -> 0 (in page 6, section 3.1). Then, figure 2 argues that collapse may not occur because empirical NC1 is not close to 0. First, how to define "NC1 -> 0". For Waterbirds, it shows NC1 could get smaller than 0.05, while for CivilComments, I think it gets very close to 0, even before the standard number of training steps. Given that every NC1 is an estimation (see my previous concern), it is not convincing that "empirical NC1 is not close to 0".

(4) For section 3.2, the authors claim Table 2 to be "Group test accuracy is not well-correlated with the minimum l2". But just from the table I cannot see it. For example, three groups in CivilComments are negative and one is positive. Group 1 of MultiNLI is notably higher than the other 5. The authors need to conduct statistical testing here (e.g., p-value).

(5) Given the above (3), (4), the overthrowing of the common hypothesis is not well-supported. On the writing side, I feel the authors are being more conservative in sections 3.1 and 3.2, e.g., "neural collapse may not occur" in section 3.1 title versus "does not occur" in the contributes bullet point 2; "is not well-correlated" in Table 2 caption versus "convergence of the LLR classifier to the maximum-margin solution is extremely slow". The certainty throughout this paper should at least be self-consistent. PLEASE BE CAUTIOUS AND DO NOT OVERCLAIM, given that this research tries to deny the current hypothesis.

(6) In section 4, this research proposes that the LLR performance is primarily due to group balance in the held-set, but how can we exclude the effect of neural collapse? Why not "group balance mitigates neural collapse so the WGA becomes better"? Some ablation is needed here. I understand section 4.1 tries to "directly control the group ratio", but this controlled group ratio could also affect neural collapse.

(7) There should be some explanation on Figure 4 why (d) MultiNLI first increases, then decreases. Is this a counter-example of the proposed group-balance hypothesis?

(8) As an audience who hasn't read any work on last-layer retraining, after 10 minutes of reading, I don't quite get Figure 5. LLR can recover optimally class-balanced WGA, so what? How does this relate to "LLR performance is primarily due to group balance in the held-out set"?

**Requested Changes:**

I think my explanation of why I feel the claims made in the submission are not well-supported is clear enough. For each weakness I've identified, either I try to provide a way to fix it, or it's straightforward how to fix it. I don't want to copy-paste and make this review super long, but if the authors have any confusions please let me know right away, and I'll get back ASAP.

---

> ### Author Response · Authors · 2026-02-16
> **Author response**
>
> We graciously thank reviewer souW for their careful reading and detailed questions. We appreciate that the reviewer recognizes our “very interesting” contributions and the risky-but-impactful nature of our research. We are also grateful to the reviewer for spending substantial time building expertise on the preliminaries and related literature to most effectively review our paper.
>
> Before we address each of the reviewer’s points, we would like to clarify a potential misunderstanding regarding the overall positioning of our paper. The reviewer mentions that we “overthrow a common hypothesis”, but we would like to soften this language a bit and not overclaim our contribution. In our experience, the commonly held view is that LLR is an exceptionally flexible method in terms of group balance -- specifically, [11, 12] showed that LLR on a held-out set with the **same data distribution** as the training set can still improve WGA, and the reason why is not fully understood. On the other hand, the hypothesis that LLR’s success is due to an interaction of neural collapse and implicit-bias behavior is **our** contribution; we have discussed it with members of the community who think it reasonable, but we would not say it is commonly held.
>
> Below, we respond to each point in the reviewer’s “Claims and Evidence” section.
>
> 1. We would like to address the points about model coverage and dataset coverage separately. For model coverage, we agree this can be improved with respect to Transformer-based architectures for the vision tasks. To this end, we have run all of our experiments with Swin Transformer (an efficient vision transformer [1]) on Waterbirds and included the results in Appendix A.4. We found that the results closely matched those of the ResNet, suggesting that the phenomena we study are agnostic to convolutional vs. Transformer-based models. The experiments for Swin Transformer on CelebA are actively running and will be included in any camera-ready version of the paper. For the last point about modern pretrained encoders (say, SigLIP [2] for vision or Gemma-2B [3] for language), these models are typically not studied in the spurious correlations literature due to a combination of compute constraints and lack of control over the pretraining data.
>
>     For dataset coverage, we hold that four datasets across two modalities meets or exceeds the standard in the spurious correlations literature, and moreover each of our datasets are well-established benchmarks in the field. Here is a short selection of datasets we considered but ultimately chose not to include: (a) ColorMNIST [4]: a common choice but too simple and small-scale, (b) Spawrious [5]: a large controlled dataset, but overlaps with Waterbirds as the spurious feature is still the image background, (c) FMOW [6]: a high-resolution satellite imagery dataset with heavy group imbalance, but contains a confounding distribution shift across the temporal axis, (d) CXR-14 [7]: an interesting medical imaging application, but lacks group annotations on the training set, (e) Yelp-Author-Style [8]: an NLP dataset where review scores are correlated with writing in the style of Shakespeare or Hemingway; however, the reviews are generated by Llama-2 and thus contain exaggerated biases which do not mimic human behavior.
>
> 2. We have to use an estimator for NC1 due to the inordinate memory requirement for computing exact values. Our Table 1 indicates that computing exact values of NC1 can take **up to 425GB of memory**; hence, it is not common in the neural collapse literature to compute NC1 for models as large as BERT-Base. Our Hutchinson estimator reduces this memory requirement to only 4MB.
>
>     In order to address your concerns regarding the accuracy of the estimator, we have added a new appendix section to our revised draft. Appendix C shows that under mild assumptions, the relative variance of our NC1 estimate scales as O(1/KN) where K is the number of random samples and N is the feature dimension. Large models benefit from this dimension dependence and require only a handful of random samples to produce stable estimates. We found that K=3 was sufficient to observe stable results on the largest models, with K=10 being more realistic for smaller models.

---

> > ### Author Response · Authors · 2026-02-16
> > **Author response**
> >
> > 3. We thank the reviewer for this valuable comment. To contextualize our comments regarding NC1 not going to 0, we would point the reviewer to Figure 6 from the original neural collapse paper [9]. The value of NC1 at the end of training in [9] is much lower than in our paper. In [9], NC1 decreases to nearly 1e-4 for several model/dataset combinations, whereas for all of our tested models and datasets NC1 remains above 1e-2. Also, in Figure 6 of [9] they show that the decay of NC1 accelerates following interpolation (i.e., when the model perfectly fits the training dataset). Despite the fact that we trained past interpolation for all models, we do not observe this behavior: NC1 remains flat from interpolation to the end of training in all of our experiments, i.e., if the current rate of change continued in Waterbirds it would take 100K+ additional iterations to reach 1e-2.
> >
> > 4. To clarify, we use the Pearson correlation coefficient where -1.0 indicates a perfect negative correlation, 0.0 indicates no correlation, and +1.0 indicates a perfect positive correlation. Thus, the phenomenon noticed by the reviewer where three groups in CivilComments are positive and one is negative **supports** our claim that group test accuracy is not well-correlated with minimum margin: all four values are close to zero, and the correlation is not even consistently positive. Likewise, no group has a correlation coefficient above 0.526 (including the MultiNLI group 1 referenced by the reviewer), indicating at best a weak-to-moderate correlation.
> >
> > 5. We appreciate the suggestions. We have revised the language (particularly in regards to neural collapse) to be self-consistent and avoid overclaiming. However, in the case of the maximum margin SVM experiments we believe that the different language across sections is actually complimentary. We reject the role of the maximum margin SVM in our hypothesis for two reasons. Firstly, we demonstrate that convergence to the SVM classifier is slow and is not close to complete after a standard number of training epochs. Secondly, the results in Table 2 show that the minimum margin of the last-layer classifier is not highly correlated to test accuracy across groups. Thus, even were the last-layer to converge quickly to the maximum margin SVM, it could still be the case that the LLR classifier would exhibit low WGA. Therefore, we do not find evidence to suggest the implicit bias of gradient descent is a major cause of LLR’s success.
> >
> > 6. Our experiments in Section 3 rigorously show that neural collapse does not occur during the standard length of training even for **imbalanced datasets**. In other words, there is no significant neural collapse behavior that group balance could mitigate. We believe this excludes neural collapse as a confounder in Section 4.
> >
> > 7. This is a great point, and we appreciate the reviewer’s careful attention to the details of the figures. We believe this is an example of **exact** group balance (i.e., MultiNLI having a 1/6 proportion of all 6 groups) not aligning with the **optimal** group balance for WGA. This is a phenomenon which has only recently been identified in the literature; the most relevant paper is [10], which shows that a slight overrepresentation of the majority group is necessary to achieve the best WGA on MultiNLI. Intuitively, this may be the case if the majority group is more complex or diverse than the minority group, and thus requires more data to achieve the same group accuracy. We have clarified that our hypothesis refers to the distance from **exact** group balance, not necessarily **optimal** group balance (though they roughly align in the other datasets).
> >
> > 8. We appreciate this comment; clarifications here will improve the accessibility of our work. Section 4, including Figure 5, is a direct response to the findings of [11, 12], who showed that LLR on a held-out set with the **same data distribution** as the training set can still improve WGA. In this case, it seems impossible that group balance could be a factor, which led [11] to term LLR a “free lunch”. Our main contribution in Section 4 is that, despite the framing of [11, 12], we show the robustness improvements of LLR **can still be attributed to group balance**. The surprising part is that the improvement in data distribution alignment is completely implicit: it is induced by the class-balancing methodology of [11] and the loss-based upweighting technique of [12].

---

> > > ### Author Response · Authors · 2026-02-16
> > > **Author response**
> > >
> > > Regarding broader impact concerns: We wholeheartedly share your concern regarding ethical usage of demographic data. As artificial intelligence becomes more broadly applicable, it is critically important to ensure one’s research does not amplify social inequities and aligns with the highest standards of responsible AI development. To this end, we have included dataset license information in Appendix B and made every effort to ensure responsible practices. For example, we use the objective hair color classification task in CelebA and discard potentially unethical annotations such as attractiveness, for which substantial scholarship exists regarding its biased nature in CelebA specifically [13, 14].
> > >
> > > [1] Liu et al. “Swin Transformer: Hierarchical Vision Transformer using Shifted Windows.” ICCV 2021.
> > >
> > > [2] Zhai et al. “Sigmoid Loss for Language Image Pre-Training.” ICCV 2023.
> > >
> > > [3] Gemma Team. “Gemma: Open Models Based on Gemini Research and Technology.” ArXiv 2024.
> > >
> > > [4] Arjovsky et al. “Invariant Risk Minimization.” ArXiv 2019.
> > >
> > > [5] Lynch et al. “Spawrious: A Benchmark for Fine Control of Spurious Correlation Biases.” ICLR 2025 Workshop on Spurious Correlations and Shortcut Learning.
> > >
> > > [6] Koh et al. “WILDS: A Benchmark of in-the-Wild Distribution Shifts.” ICML 2021.
> > >
> > > [7] Wang et al. “ChestX-ray8: Hospital-scale Chest X-ray Database and Benchmarks on Weakly-Supervised Classification and Localization of Common Thorax Diseases.” CVPR 2017.
> > >
> > > [8] Zhou et al. “Navigating the Shortcut Maze: A Comprehensive Analysis of Shortcut Learning in Text Classification by Language Models.” EMNLP 2024.
> > >
> > > [9] Papyan et al. “Prevalence of neural collapse during the terminal phase of deep learning training.” PNAS 2020.
> > >
> > > [10] Qiao et al. “Group-robust Sample Reweighting for Subpopulation Shifts via Influence Functions.” ICLR 2025.
> > >
> > > [11] LaBonte et al. “Towards last-layer retraining for group robustness with fewer annotations.” NeurIPS 2023.
> > >
> > > [12] Qiu et al. “Simple and Fast Group Robustness by Automatic Feature Reweighting.” ICML 2023.
> > >
> > > [13] Prabhu et al. “Covering up bias with Markov blankets: A post-hoc cure for attribute prior avoidance”. ICML 2019 Workshop on Invertible Neural Networks and Normalizing Flows.
> > >
> > > [14] Rajabi et al. “Through a fair looking-glass: mitigating bias in image datasets.” HCII 2023.

---

> ### Comment · Reviewer_souW · 2026-03-01
> **Thank you for your clarifications.**
>
> TL; DR: Clear on model coverage; not sure on data coverage; and further comments on estimator;
> clear on NC not ->0;
> No obviously ethical concern identified
>
> -----------------------------------------------
> I appreciate the authors' clarifications.
>
> > (1) a) model coverage
>
> I believe I'm now clear with this. The extended models help establish a more rigorous evaluation.
>
> > (1) b) dataset coverage
>
> I'm actually not from the spurious correlation community. I'm not sure if the datasets already exceed the standard, so I have to trust the Action Editor's view on this.
>
> > (2) study on proposed estimator
>
> I appreciate the extended theoretical analysis. Is it possible to do some small-scale experiments to study the proposed estimator on a smaller data/model?
>
> > (3) how small is NC1 -> 0?
>
> Your empirical NC1 is indeed much larger than [9]. I would suggest adding a comment on this in the paper so that future readers won't confuse.
>
> > (4) Table 2 statistical testing
>
> I'm clear on this.
>
> > (5) Overall claim rigor.
>
> I'm clear on this and thank the authors for the re-writing.
>
> > (6) - (8)
>
> I appreciate the clarifications from the authors. Please incorporate them into the next version.
>
> > broader impact
>
> The authors are able to further defend their ethical impact. I do not see any notable ethical concerns at this time.

---

### Review · Reviewer_aaL8 · 2026-02-02

**Summary Of Contributions:**

This paper investigates the underlying mechanisms of Last-Layer Retraining (LLR) for improving group robustness. The authors first examine the hypothesis that LLR works by mitigating Neural Collapse. Through empirical analysis, they demonstrate that the degree of Neural Collapse does not correlate with the robustness improvements observed in LLR. To facilitate this analysis, the paper proposes a memory-efficient algorithm based on the Hutchinson trace estimator. This method significantly reduces the memory cost for computing the $\mathcal{NC}_1$ metric on large-scale models. Furthermore, the authors provide strong evidence that the effectiveness of LLR is primarily driven by better group balance in the held-out set. Finally, the study analyzes recent state-of-the-art algorithms, specifically CB-LLR and AFR, and shows that these methods function by performing implicit group balancing.

**Audience:**

Yes

**Audience Explanation:**

- This paper explains the mechanism of LLR. The findings identify the cause of performance improvements. The authors link these improvements to group balance in the held-out set.

- The paper introduces an algorithm for trace estimation with the Hutchinson estimator. This approach reduces memory usage. It enables the calculation of metrics on models like BERT.

**Claims And Evidence:**

Yes

**Claims Explanation:**

- This paper claims that the NC phenomenon does not explain the success of LLR. They support this claim with empirical testing in Section 3. They measure the correlation between the $\mathcal{NC}_1$ and worst-group accuracy across different training settings. The data clearly shows that improvements in robustness do not correspond to changes in Neural Collapse.
- This paper claims that the group balance of the held-out set is the primary driver of LLR’s performance. The authors validate this in Section 4 through extensive experimentation. They vary the class balance of the held-out set on four standard benchmarks (Waterbirds, CelebA, CivilComments, MultiNLI). The results demonstrate a strong and consistent correlation between the balance of the held-out set and the final worst-group accuracy.
- This paper claims their proposed algorithm for estimating Neural Collapse is memory-efficient. The paper provides a direct comparison of memory usage between the standard method and their proposed Hutchinson trace estimator. The reported reduction from hundreds of GiBs to a few MiBs constitutes accurate and objective evidence.

**Requested Changes:**

This paper introduces a memory-efficient algorithm using the Hutchinson trace estimator. Please discuss the variance of this estimator. The authors should clarify how the number of random vectors affects the accuracy of the $\mathcal{NC}_1$ calculation. It would be helpful to state the minimum number of vectors required for stable results.

---

> ### Author Response · Authors · 2026-02-16
> **Author response**
>
> We warmly thank Reviewer aaL8 for their detailed comments. We appreciate that the reviewer recognizes our “extensive experimentation” which elucidates the “strong and consistent correlation” between group balance and the worst-group accuracy attained by LLR. We agree that our proposed Hutchinson estimator “significantly reduces the memory cost” for computing neural collapse metrics on 100M+ parameter scale models, and we anticipate that the community will find this technique to be of independent interest.
>
> Regarding the variance of the Hutchinson estimator, please see Appendix C in our revised draft. We include a detailed analysis of the variance of our NC1 estimate. Importantly, we show that under mild assumptions, the relative variance of our NC1 estimate scales as O(1/KN) where K is the number of random samples and N is the feature dimension. Therefore, the number of random samples required to achieve stable results decreases in the feature dimension. This is desirable as large models are the expected application for the NC1 estimator and they benefit from this dimension dependence. We found that K=3 was sufficient to observe stable results on the largest models, with K=10 being more realistic for smaller models.

---

### Decision · Action_Editor_iV2N · 2026-03-28

**Recommendation:** Accept with minor revision

**Additional Comments:**

- Dataset diversity and coverage. The current evaluation appears limited in scope, with datasets primarily drawn from CV and NLP domains. Please comment on whether this level of diversity is sufficient to support the paper’s claims, or if additional justification (or experiments) is needed.

- The manuscript would benefit from substantial revisions to improve organization, clarity, and readability. In particular, the authors should ensure that the final version is polished and easily understandable to first-time readers, without requiring reliance on the rebuttal for clarification.

- There is another related paper about last-layer retraining [[1](https://arxiv.org/abs/2204.09583)], which uses a sample-splitting procedure and may be worth discussing in the related work.

**Audience:**

Yes

**Audience Explanation:**

The paper would be of interest to audiences who work on spurious correlations and imbalanced classification.

**Claims And Evidence:**

Yes

**Claims Explanation:**

This paper examined last-layer retraining methods to mitigate spurious correlations. All reviewers agree that the paper is well written and that the ideas are interesting.

Several reviewers had major concerns about the core claims and the overall structure of the paper. After a productive discussion, these concerns were addressed, and reviewers reached a consensus. As a result, I would like to concur with their recommendations and recommend acceptance.

However, there are still numerous concerns from the reviewers that should be incorporated into the final version. Please make sure to go through the reviewers' comments carefully (See also several additional comments below).

---

> ### Author Response · Authors · 2026-05-10
>
> Dear AE,
>
> Thank you very much for your consideration of our submission. We have uploaded the final version of our work and would like to briefly address your additional comments:
>
> 1. We believe our dataset diversity and coverage is sufficient to support our claims; we hold that four datasets across two modalities meets or exceeds the standard in the spurious correlations literature. For further discussion, including a selection of datasets we considered but ultimately chose not to include, please see our response (1) to Reviewer souW. Overall, we believe our selection of four datasets represent among the most rigorous and widely used benchmarks in the field, making our insights broadly relevant across a range of settings and modalities susceptible to spurious correlations.
>
> 2. We appreciate your attention to the quality and accessibility of our writing. We have taken the time to substantially revise our final version, and we believe it is now more understandable to both first-time readers and experts.
>
> 3. Thank you for the interesting reference; we have included it in the relevant section of our related work.